# Boosting Adversarial Transferability with Spatial Adversarial Alignment

**Zhaoyu Chen**[1*]   **Haijing Guo**[2,3*]   **Kaixun Jiang**[1]   **Jiyuan Fu**[2]   **Xinyu Zhou**[2]
**Dingkang Yang**[1]   **Hao Tang**[4]   **Bo Li**[5]   **Wenqiang Zhang**[1,2†]

[1]College of Intelligent Robotics and Advanced Manufacturing, Fudan University
[2]College of Computer Science and Artificial Intelligence, Fudan University
[3]China Southern Power Grid Artificial Intelligence Technology Co., Ltd.
[4]School of Computer Science, Peking University     [5]vivo Mobile Communication Co., Ltd.
{zhaoyuchen20, wqzhang}@fudan.edu.cn, hjguo22@m.fudan.edu.cn

## Abstract

Deep neural networks are vulnerable to adversarial examples that exhibit transferability across various models. Numerous approaches are proposed to enhance the transferability of adversarial examples, including advanced optimization, data augmentation, and model modifications. However, these methods still show limited transferability, particularly in cross-architecture scenarios, such as from CNN to ViT. To achieve high transferability, we propose a technique termed Spatial Adversarial Alignment (SAA), which employs an alignment loss and leverages a witness model to fine-tune the surrogate model. Specifically, SAA consists of two key parts: spatial-aware alignment and adversarial-aware alignment. First, we minimize the divergences of features between the two models in both global and local regions, facilitating spatial alignment. Second, we introduce a self-adversarial strategy that leverages adversarial examples to impose further constraints, aligning features from an adversarial perspective. Through this alignment, the surrogate model is trained to concentrate on the common features extracted by the witness model. This facilitates adversarial attacks on these shared features, thereby yielding perturbations that exhibit enhanced transferability. Extensive experiments on various architectures on ImageNet show that aligned surrogate models based on SAA can provide higher transferable adversarial examples, especially in cross-architecture attacks.

## 1   Introduction

Deep neural networks (DNNs) have been successfully and extensively deployed across security-sensitive applications [32], including autonomous driving [63, 68, 67], facial verification [70, 69, 47, 9], and video surveillance [23, 72, 24, 20]. However, DNNs exhibit considerable vulnerability to adversarial examples [18, 39, 16, 6, 7, 5, 4, 30, 31, 33], where imperceptible perturbations are introduced into natural images, leading models to produce incorrect predictions. In real-world applications, DNNs are typically concealed from user access, necessitating adversaries to generate adversarial examples within a black-box setting, where no knowledge of the target model's parameters or architecture is available. Adversarial transferability [13, 26, 51] plays a crucial role in black-box settings as it allows adversaries to effectively compromise target models by employing adversarial examples generated on surrogate models. In black-box settings, adversarial transferability plays a crucial role, which enables adversaries to leverage adversarial examples crafted on surrogate models

---

[*]indicates equal contributions.
[†]indicates corresponding authors.

39th Conference on Neural Information Processing Systems (NeurIPS 2025).

to effectively attack target models. Thus, generating highly transferable adversarial examples is instrumental in uncovering and understanding the vulnerabilities within DNNs, drawing substantial attention in recent research.

Cross-model transferability has been extensively studied for CNNs [13, 61, 14]. Highly transferable adversarial examples are usually based on advanced optimization [13, 29, 51] and data augmentation [61, 14, 37]. The principle is to alleviate the overfitting of adversarial examples on surrogate models, determining whether the attack can be successfully transferred to the target models. In addition, some model modification methods [59, 21], such as amplifying the gradient on skip connections (the structure in ResNet [22]), can also improve transferability. However, few works explore adversarial transferability on Vision Transformer (ViT) [15] and the performance of existing work extending CNN to ViT is poor due to significant structural differences. Specifically, ViT flattens the image into a sequence of patch tokens and employs multi-head self-attention to capture global relationships among the patches. In contrast, CNNs typically consist of stacked convolutional layers that learn feature relationships progressively through downsampling. Therefore, [58] first empirically analyzes the structure of ViT and propose PNA and PatchOut [58], but there is still much room for improvement in cross-architecture transferability.

In this paper, we argue that unique structural features are critical to cross-architecture adversarial transferability. Given a dataset, various models tend to exhibit analogous decision boundaries [34], arising from their ability to learn similar features. If we can obtain a surrogate model whose features are similar to those of models with different architectures, then the resulting adversarial perturbation can be transferable across different models. A recent technique known as Model Alignment (MA) [38] employs an alignment loss to minimize prediction divergences between surrogate models and witness models, thereby indirectly facilitating the extraction of features that are similarly represented by the witness model. However, directly applying MA to black-box attacks may lead to the degradation of cross-architecture transferability. The main reasons are: **(i)** Features are not aligned in space [58]. MA only uses the final prediction of the model, but in fact, the spatial features of ViT and CNN are different [71, 57]. It is difficult to directly constrain the similarity of features only by the final logits. **(ii)** Features are not aligned from the perspective of adversarial features. In addition to the features of clean images, the features of adversarial examples also have similarities across different models and need to be considered.

To overcome these challenges and enhance transferability, we propose a technique called Spatial Adversarial Alignment (SAA), which utilizes an alignment loss from the perspective of spatial and adversarial features and incorporates a witness model to refine the surrogate model. SAA consists of two key parts: spatial-aware alignment and adversarial-aware alignment. In the spatial-aware alignment, in addition to aligning on the final global features, we also focus on the features of local regions. We make local features of CNNs by position to align ViTs' embeddings at the same position. In the adversarial-aware alignment, we introduce a self-adversarial strategy, which constructs adversarial examples so that the model can learn the differences between different architectures in adversarial features, thereby enabling the model to further capture more common features. Aligned surrogate models by SAA provide promising adversarial transferability and can be seamlessly integrated with existing transfer attacks. In addition, we further summarize the empirical guidance for the selection of surrogate and witness models in SAA. Our contributions can be summarized as follows:

• We reveal for the first time the importance of spatial and adversarial features for cross-architecture transferability, which supports alignment with different models.

• We propose Spatial Adversarial Alignment (SAA), which leverages a witness model to fine-tune the surrogate model via spatial-aware and adversarial-aware alignment to generate highly transferable adversarial examples. In addition, we further summarize the empirical guidance for the model selection in SAA.

• Experiments on 6 CNNs and 4 ViTs show that SAA has state-of-the-art adversarial transferability, especially in cross-model transferability. Compared with MA, on ResNet50, the transferability from CNN to ViT is improved by 25.5-39.1%.

## 2 Methodology

### 2.1 Preliminaries

In this paper, we focus on the image classification task on DNNs. Let $f_\theta(\cdot)$ represent a DNN-based classifier with different network parameters $\theta$. We denote the clean image as $x$ and its corresponding ground-truth label as $y$. Following [13, 61, 14], we evaluate the adversarial transferability under untargeted adversarial attacks with $l_\infty$ norm. Therefore, the goal of transfer attacks is to add an adversarial perturbation to the clean image $x$ based on the information of the surrogate model $f_{\theta_s}(\cdot)$ to obtain the adversarial example $x_{adv}$ [18], so that $f_{\theta_s}(x_{adv}) \neq y$ subject to the constraint that $||x_{adv} - x||_\infty \leq \epsilon$. In the black-box setting, no information about the target model—such as its architecture, weights, or gradients—is accessible. Therefore, adversarial examples are generated solely by utilizing a surrogate model $f_{\theta_s}(\cdot)$, leveraging their transferability to deceive the target model $f_{\theta_t}(\cdot)$.

### 2.2 Spatial Adversarial Alignment (SAA)

Spatial Adversarial Alignment (SAA) employs an alignment loss tailored to both spatial and adversarial feature perspectives, incorporating a witness model to fine-tune the surrogate model. SAA aims to adjust the surrogate model to extract features closely aligned with those of the witness model, capturing both spatial and adversarial features shared across models. As shown in Figure 1, SAA consists of two parts, namely spatial-aware alignment and adversarial-aware alignment.

**Spatial-aware Alignment.** The purpose of spatial-aware alignment is to make the surrogate and witness models more consistent in the feature space. Naturally, the most intuitive approach to aligning the feature distributions of two models is to minimize the distance between their final outputs [38]. However, when the models exhibit significant architectural differences, ensuring output similarity alone is insufficient to achieve alignment in intermediate features. In black-box attacks, where the details of the target model's architecture are unknown, this issue becomes more pronounced. Taking the challenging scenario of CNN to ViT as an example, their intermediate layer features differ substantially in semantic levels [71]. This discrepancy arises primarily from differences in receptive fields, stacking methodologies, and normalization techniques between CNNs and ViTs. Therefore, relying solely on output alignment for model fine-tuning indirectly captures some common features, but this approach can, in certain cases, result in degraded transferability, as observed in methods like Model Alignment (MA) [38].

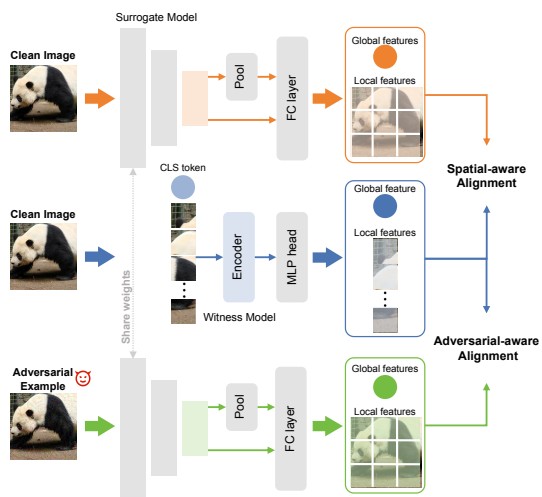

Figure 1: Spatial Adversarial Alignment (SAA) consists of two parts: spatial-aware alignment and adversarial-aware alignment. Initially, we aim to minimize the feature divergences between the two models across both global and local regions, thereby promoting spatial alignment. Subsequently, we introduce a self-adversarial strategy that utilizes adversarial examples to impose additional constraints, aligning the adversarial features.

Therefore, in addition to aligning on the final global features, we also need to focus on the features of local regions. For ease of understanding, we define the global features $f_\theta(x)$ as the logits of the model corresponding to the input $x$. For CNNs, it is the output of features by the last layer. For ViTs, it refers to the final embedding of the `[CLS]` token after the MLP block. First, we perform alignment at the global feature level by defining an alignment loss between the surrogate model and witness model at the output layer:

$$\mathcal{L}_{global}(x; \theta_s) = D_{\mathrm{KL}}(f_{\theta_s}(x), f_{\theta_w}(x)), \tag{1}$$

where $D_{\mathrm{KL}}$ measures the feature divergence with Kullback-Leibler (KL) divergence.

Next, we align the models at the local feature level. Here, we define $z_\theta(x)$ as local features. Each spatial position $(h, w)$ within this feature map is treated as a distinct local region and the feature for each local region is $z_\theta^{[q]}(x)$, where $q = \{1, 2, ..., H \times W\}$. Since every local feature is embedded with the corresponding sub-image and position information, regarding them as spatial dense predictions is reasonable. Let $z_{\theta_s}^{[q]}(x)$ and $z_{\theta_w}^{[q]}(x)$ denote the local features associated with each local region $q$ for the surrogate model and witness model, respectively. For CNNs, $z_\theta(x)^{B \times C \times H \times W}$ represents the logits generated by the final convolutional layer and then pass the final MLP. For ViTs, $z_\theta(x)^{B \times C' \times H \times W\,3}$ is the embeddings of patch tokens after passing through the last MLP except for the [CLS] token, where each patch token corresponds to a specific spatial region in the input. Next, we argue pseudo-labels better aggregate local information, so we compute pseudo-labels of the local region $q$ and denote this pseudo-label as $\hat{y}_{\theta_w}^{[q]}$, which is obtained by taking the $\arg\max$ over the logits after the last MLP of the witness model: $\hat{y}_{\theta_w}^{[q]} = \arg\max(z_{\theta_w}^{[q]}(x))$. Then, we use this pseudo-label to supervise the learning of the local feature of the surrogate model. To achieve local alignment, we minimize the divergence of corresponding local regions' logits and the local alignment loss is expressed as:

$$\mathcal{L}_{local}(x; \theta_s) = \frac{1}{HW} \sum_{q=1}^{HW} D_{\text{CE}}(z_{\theta_s}^{[q]}(x), \hat{y}_{\theta_w}^{[q]}), \tag{2}$$

where $D_{CE}$ is the cross-entropy loss. Therefore, the spatial-aware alignment loss is expressed as:

$$\mathcal{L}_{SA}(x; \theta_s) = \mathcal{L}_{global}(x; \theta_s) + \gamma \cdot \mathcal{L}_{local}(x; \theta_s), \tag{3}$$

where $\gamma$ is the spatial factor. By minimizing this spatial-aware alignment loss, we encourage the surrogate model to produce features in both global and local regions that are consistent with those of the witness model, even across different architectures.

**Adversarial-aware Alignment.** The relationship between features and adversarial vulnerability is highly significant. Some hypotheses [64, 43] propose that adversarial examples possess distinct feature distributions compared to normal examples, which may inherently predispose models to adversarial vulnerability—a notion supported by several studies [1, 37]. Beyond normal examples, learning adversarial features may offer a way to capture shared features between surrogate models and witness models. Furthermore, [14] suggests that models trained with adversarial examples focus on more discriminative regions within images, displaying feature recognition patterns distinct from those of normally trained models. Thus, adversarial examples play a crucial role in achieving model alignment.

In our adversarial-aware alignment, we introduce a self-adversarial strategy that constructs adversarial examples of the surrogate model to enable the model to discern architectural differences in adversarial features effectively. Specifically, we leverage the gradients to iteratively generate adversarial examples under the supervision of the global features of the witness model. Assuming $x_{adv}^{(0)} = x$, we define the adversarial example $x_{adv}^{(t+1)}$ of the surrogate model as:

$$x_{adv}^{(t+1)} = \Pi_{x,\epsilon} \left( x_{adv}^{(t)} + \alpha \cdot \text{sign} \left( \nabla_x D_{\text{KL}} \left( f_{\theta_s}(x_{adv}^{(t)}), f_{\theta_w}(x) \right) \right) \right), \tag{4}$$

where $D_{\text{KL}}$ denotes the KL divergence, $x_{adv}^{(t)}$ denotes the adversarial example at iteration $t$, $\alpha$ is the step size, and $\Pi_\epsilon$ projects the adversarial example onto an $\epsilon$-bounded neighborhood around the original input $x$.

Once the adversarial example $x_{adv}$ is generated, we also perform adversarial-aware alignment on the adversarial examples from global and local features to further align the surrogate and witness models. The loss of the adversarial-aware alignment is expressed as:

$$\mathcal{L}_{\mathcal{A}\mathcal{A}}(x_{adv}; \theta_s) = \mathcal{L}_{global}(x_{adv}; \theta_s) + \omega \cdot \mathcal{L}_{local}(x_{adv}; \theta_s), \tag{5}$$

where $\omega$ is the adversarial factor.

**Optimization.** Combining spatial-aware and adversarial-aware alignment, the final optimization goal of spatial-adversarial alignment is:

$$\mathcal{L}_{SAA}(x; \theta_s) = \mathcal{L}_{SA}(x; \theta_s) + \kappa \cdot \mathcal{L}_{AA}(x_{adv}; \theta_s), \tag{6}$$

---

[3]Generally, ViT's patch embeddings $z(x)$ is $(B, N, C')$ by default. We first transform it to $(B, C', H', W')$, where $N = H' \times W'$. Then, we perform an adaptive pooling operation to transform it to $(B, C', H, W)$.

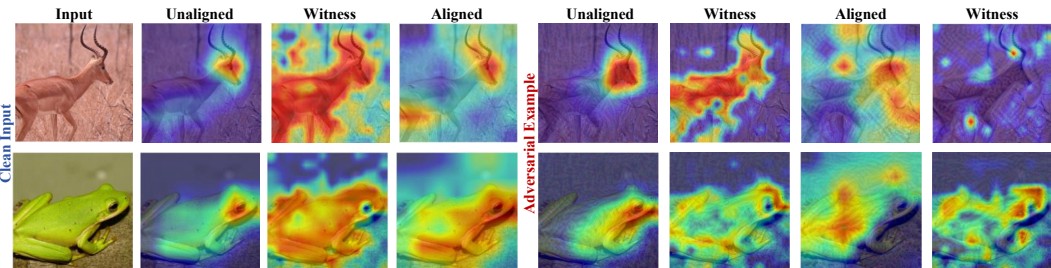

Figure 2: Grad-CAM visualizations comparing the feature distribution of unaligned and aligned surrogate models (Res50) on clean inputs and adversarial examples (generated by SSA-DI-TI-MI).

where $\kappa$ is the alignment factor to balance the two alignments. If not otherwise stated, we define $\gamma = 0.2$, $\omega = 0.02$, and $\kappa = 0.02$ in this paper.

Spatial-adversarial alignment facilitates the alignment of the surrogate model with the witness model to further improve the adversarial transferability. The parameter update rule for the surrogate model, based on stochastic gradient descent (SGD), can be expressed as follows:

$$\theta_s^{(t+1)} = \theta_s^{(t)} - \eta \cdot \frac{1}{|\mathcal{B}|} \sum_{x \in \mathcal{B}} \nabla_{\theta_s^{(t)}} \mathcal{L}_{SAA}(x; \theta_s), \tag{7}$$

where $t$ is the epoch, $\eta$ is the learning rate and $\mathcal{B}$ means the mini-batch samples. Please refer to **Algorithm** 1 for the detailed loss calculation process.

## 2.3 A Close Look at SAA

To verify whether SAA significantly improves the spatial and adversarial features after model alignment, we conduct quantitative and qualitative analyses based on the models before and after alignment. We randomly sample 100 images from ImageNet val and then compute the cosine similarity between the global features of the surrogate models before and after applying SAA with different witness models. Table 1 shows that, whether for clean images or adversarial examples (generated by SSA-DI-TI-MI), the feature similarity improves after alignment. Notably, when the surrogate model is ViT-B, the improvement in similarity is even more pronounced. In addition, the feature gap between Res50 and Swin-B is indeed significant, but even so, SAA achieves a $\frac{0.0551 - 0.0369}{0.0369} = 49.3\%$ improvement in similarity. This result suggests that, after applying SAA, the aligned surrogate models effectively capture features shared with the witness model, providing strong evidence of the alignment's success.

Then, we use Grad-CAM [45]'s heatmaps to simulate the feature distribution of the model, as shown as Figure 2. For the **clean inputs** (first four cols), the heatmaps generated by the unaligned surrogate model (2-nd col) primarily focus on local regions of the object. In contrast, the aligned surrogate model (4-th col) heatmaps demonstrate more diffuse attention spread across the entire object, similar to that of the witness model (ViT-B, 3-rd col), which shows aligned surrogate models learn the common spatial features. For **adversarial examples** (last four cols), the 5-th and 7-th cols display the heatmaps of adversarial examples

Table 1: Cosine similarity of global features of surrogate models.

| Surrogate | Witness | Clean | | Adv | |
|---|---|---|---|---|---|
| | | Unaligned | Aligned | Unaligned | Aligned |
| Res50 | Res50 | **1.0000** | 0.9949 | **1.0000** | 0.9922 |
| | DN121 | 0.0573 | **0.1153** | 0.0700 | **0.1328** |
| | ViT-B | 0.0533 | **0.1408** | 0.0452 | **0.1191** |
| | Swin-B | 0.0352 | **0.0448** | 0.0369 | **0.0551** |
| ViT-B | Res50 | 0.0566 | **0.1323** | 0.0672 | **0.1544** |
| | DN121 | 0.4016 | **0.6278** | 0.4121 | **0.6551** |
| | ViT-B | **1.0000** | 0.9706 | **1.0000** | 0.9728 |
| | Swin-B | 0.3058 | **0.5115** | 0.3169 | **0.4257** |

generated by the unaligned and aligned surrogate models, respectively. The 6-th and 8-th show the witness model's responses to these adversarial examples. Notably, the adversarial examples generated by the unaligned surrogate model fail to effectively transfer to the witness model (6-th col) due to still focusing on the target subject, indicating limited cross-model transferability. In contrast, adversarial examples generated by the aligned surrogate model (7-th col) successfully transferred to the witness model (8-th col) as the features are spread out, demonstrating enhanced cross-model transferability achieved through SAA.

# 3 Experiments

## 3.1 Experimental Setup

**Datasets.** Our experiments utilize the ImageNet-compatible dataset [27], a widely adopted subset containing 1,000 images from the ImageNet validation set [11]. This dataset is commonly used in adversarial robustness studies, such as those in [61, 13, 14].

**Models.** To assess the adversarial transferability of different network architectures, we focus on convolutional neural networks (CNNs) and vision transformers (ViTs) as the target models. For CNNs, we select the typically trained ResNet-18 (Res18), ResNet-50 (Res50) and ResNet-101 (Res101) [22], VGG-19 [46], DenseNet-121 (DN121) [25], and Inception-v3 (Inc-v3) [48]. For ViTs, we evaluate the Vision Transformer (ViT-B) [15], Swin Transformer (Swin-B) [35], Pyramid Vision Transformer (PVT-v2) [52], and MobileViT-s (MobViT) [40].

**Metric.** Adversarial transferability is quantified by calculating the average attack success rate (Avg. ASR, %) across target models (excluding the surrogate model), with a higher success rate signifying enhanced transferability. In the paper, 'n/a' is the baseline and defined as the average attack success rate obtained by generating adversarial examples using the surrogate model without any alignment.

**Implementation Details.** In our experiments, we select the MI [13] attack as the baseline for generating adversarial examples with high transferability, as it is widely recognized within the field of adversarial transferability [14, 61, 53, 37, 54, 51, 29, 56, 59, 58]. For MI, we set the perturbation magnitude $\epsilon = 16$ [13, 18], perform 10 iterations, with a step size of $\frac{16}{10} = 1.6$, and use a momentum $\mu = 1$. During the Spatial Adversarial Alignment, all surrogate models are fine-tuned for 1 epoch using stochastic gradient descent (SGD) with a momentum of 0.9, and no learning rate adjustments are applied. It is important to note that no additional data is used for fine-tuning, as it relies solely on the same training data used for both the surrogate and witness models. The number of adversarial examples generated by SAA is in a $1 : 1$ ratio with the training dataset. The settings for Model Alignment (MA) [38] are consistent with the parameters specified in the original paper.

## 3.2 Performance Comparison

**Performance comparison with alignment methods.** We first compare with existing alignment methods [38], where adversarial examples are generated based on MI [13]. Table 2 illustrates the performance difference between MA and SAA in terms of adversarial transferability, with SAA demonstrating a significant advantage over MA. For instance, when the surrogate model is Res50, and the witness model is also Res50, SAA achieves a 16.6% improvement in average ASR over the original surrogate model, compared to a modest 3.6% improvement with MA. This highlights that SAA, even without introducing additional information, enhances adversarial transferability through the alignment of adversarial features. Furthermore, when the witness models are DN121, ViT-B, and Swin-B, SAA outperforms MA by 11.9%, 10.4%, and 13.1%, respectively. In addition to the remarkable adversarial transferability that SAA provides, we make two other key observations: **(i)** MA only considers global features, which makes it difficult to align features between models with large differences, which may lead to a decrease in transferability. When the surrogate model is DN121 and the witness model is Swin-B, the ASR of Vit-B, Swin-B, PVT-v2, and MobViT is not as good as the origin DN121, which shows that relying solely on global features for alignment is not enough, and can only achieve poor transferability. **(ii)** SAA provides strong cross-architecture transferability. When the surrogate model is Res50, and the witness models are Res50, DN121, ViT-B, and Swin-B, the transferability of SAA on ViTs is improved by 39.06%, 31.29%, 25.51%, and 37.74% respectively compared with MA itself, and it also has high transferability between CNNs.

**Performance comparison with transfer attacks.** Aligned surrogate models by SAA have great potential for adversarial transferability, so existing transfer attacks such as advanced optimization and data augmentation can further improve transferability. Here, we choose Res50 as the surrogate model and ViT-B as the witness model, and superimpose them with MI [13], NI [29], GI [51], DI [61], TI [14] and SSA [37] to evaluate the transferability, as shown in Table 3. Taking GI and SSA as examples, the transferability of the model after SAA is improved by 21.7% and 8.4% respectively, compared with the origin surrogate model, which is a very significant improvement. When multiple attacks are integrated, such as SSA-DI-TI-MI, SAA further enhances the transferability by 2.0%, achieving an impressive 88.2% ASR, which closes white-box attacks' performance. This indicates

Table 2: Comparison of adversarial transferability on different alignment methods.

| Surrogate | Witness | Attack | Target Model | | | | | | | | | | Avg. ASR (%) |
| | | | CNNs | | | | | | ViTs | | | | |
| | | | Res18 | Res50 | Res101 | VGG19 | DN121 | Inc-v3 | ViT-B | Swin-B | PVT-v2 | MobViT | |
|---|---|---|---|---|---|---|---|---|---|---|---|---|---|
| **Res50** | n/a | MI | 57.7 | 99.9 | 58.1 | 54.2 | 55.1 | 39.0 | 9.4 | 33.0 | 38.0 | 35.7 | 42.2 |
| | Res50 | MA | 60.4 | 99.8 | 56.4 | 60.3 | 67.3 | 44.3 | 12.1 | 35.6 | 37.2 | 39.0 | 45.8 |
| | | SAA | 77.5 | 100.0 | 71.7 | 72.0 | 77.0 | 58.7 | 19.9 | 47.8 | 51.8 | 52.8 | 58.8 |
| | DN121 | MA | 83.1 | 96.7 | 75.8 | 82.3 | 87.1 | 64.0 | 19.8 | 49.6 | 54.5 | 59.2 | 63.9 |
| | | SAA | 92.4 | 98.6 | 87.1 | 90.3 | 94.6 | 77.8 | 30.5 | 64.2 | 69.2 | 76.5 | 75.8 |
| | ViT-B | MA | 74.2 | 99.2 | 63.5 | 69.3 | 72.8 | 51.5 | 18.5 | 41.5 | 42.7 | 47.4 | 53.5 |
| | | SAA | 84.1 | 99.6 | 74.7 | 80.3 | 81.8 | 65.7 | 24.3 | 48.7 | 52.9 | 62.5 | 63.9 |
| | Swin-B | MA | 64.7 | 90.4 | 50.6 | 61.2 | 61.6 | 42.3 | 10.7 | 33.6 | 36.4 | 38.8 | 44.4 |
| | | SAA | 79.7 | 95.7 | 66.4 | 74.1 | 75.9 | 56.9 | 19.6 | 41.9 | 47.9 | 55.2 | 57.5 |
| **DN121** | n/a | MI | 84.4 | 69.6 | 54.8 | 76.6 | 100.0 | 56.5 | 16.2 | 42.1 | 43.9 | 52.6 | 58.6 |
| | Res50 | MA | 88.8 | 65.8 | 49.8 | 78.3 | **100.0** | 55.7 | 14.9 | 40.0 | 38.2 | 53.0 | 57.6 |
| | | SAA | 95.7 | 80.3 | 70.9 | 87.7 | 99.6 | 78.2 | 31.4 | 55.0 | 56.6 | 72.1 | 71.9 |
| | DN121 | MA | 79.3 | 69.5 | 54.9 | 78.8 | **100.0** | 55.4 | 12.3 | 39.9 | 41.7 | 52.4 | 57.2 |
| | | SAA | 90.1 | 81.4 | 71.5 | 87.9 | 99.9 | 75.0 | 25.3 | 54.9 | 56.4 | 69.7 | 70.1 |
| | ViT-B | MA | 89.6 | 80.6 | 71.1 | 88.1 | **100.0** | 70.0 | 22.4 | 53.9 | 58.4 | 68.7 | 69.1 |
| | | SAA | 94.0 | 83.6 | 75.2 | 90.3 | 99.8 | 81.7 | 27.2 | 58.0 | 59.1 | 80.3 | 74.0 |
| | Swin-B | MA | 88.3 | 63.9 | 49.2 | 77.8 | **100.0** | 54.4 | 14.5 | 38.1 | 37.5 | 52.2 | 56.9 |
| | | SAA | 94.8 | 80.0 | 69.1 | 88.0 | 99.7 | 74.4 | 26.3 | 50.2 | 51.6 | 71.2 | 69.5 |
| **ViT-B** | n/a | MI | 48.6 | 41.0 | 31.9 | 58.1 | 50.8 | 44.4 | 100.0 | 58.1 | 44.7 | 47.4 | 53.8 |
| | Res50 | MA | 65.8 | 51.9 | 43.0 | 66.9 | 60.0 | 51.5 | 99.7 | 66.0 | 57.3 | 57.6 | 63.1 |
| | | SAA | 87.3 | 80.6 | 75.9 | 86.1 | 87.5 | 78.0 | 99.9 | 91.0 | 85.4 | 85.2 | 86.3 |
| | DN121 | MA | 88.9 | 73.6 | 67.6 | 89.1 | 87.8 | 74.9 | **100.0** | 84.9 | 78.3 | 82.7 | 83.8 |
| | | SAA | 94.7 | 86.8 | 81.3 | 93.8 | 94.0 | 86.6 | 99.8 | 91.3 | 87.2 | 90.7 | 91.0 |
| | ViT-B | MA | 54.3 | 37.2 | 29.2 | 56.0 | 47.9 | 42.4 | **100.0** | 52.8 | 42.3 | 47.4 | 52.5 |
| | | SAA | 62.7 | 46.9 | 40.9 | 64.2 | 57.5 | 52.1 | 100.0 | 59.5 | 52.6 | 58.8 | 60.9 |
| | Swin-B | MA | 66.6 | 46.0 | 38.3 | 67.5 | 57.6 | 50.8 | 99.7 | 65.7 | 53.6 | 58.2 | 62.0 |
| | | SAA | 83.0 | 69.2 | 65.6 | 81.2 | 78.1 | 74.2 | 99.3 | 84.3 | 76.0 | 78.3 | 80.0 |
| **Swin-B** | n/a | MI | 48.2 | 31.3 | 20.0 | 49.3 | 34.3 | 29.7 | 13.9 | 100.0 | 45.4 | 41.4 | 42.5 |
| | Res50 | MA | 54.8 | 50.9 | 36.5 | 61.8 | 49.3 | 40.1 | 32.4 | **100.0** | 68.5 | 55.6 | 55.4 |
| | | SAA | 90.3 | 85.3 | 77.9 | 92.0 | 88.4 | 76.4 | 64.7 | 99.9 | 92.4 | 89.0 | 85.7 |
| | DN121 | MA | 77.3 | 71.4 | 60.0 | 83.9 | 76.4 | 60.7 | 49.0 | **100.0** | 85.9 | 80.9 | 74.9 |
| | | SAA | 94.9 | 91.7 | 85.8 | 96.5 | 94.7 | 85.6 | 74.3 | 100.0 | 95.5 | 95.4 | 91.4 |
| | ViT-B | MA | 62.6 | 52.8 | 40.0 | 66.2 | 55.3 | 45.6 | 34.0 | **100.0** | 70.1 | 61.8 | 59.5 |
| | | SAA | 84.7 | 80.0 | 74.2 | 90.5 | 85.5 | 75.7 | 70.3 | 100.0 | 91.9 | 88.6 | 84.6 |
| | Swin-B | MA | 62.6 | 52.8 | 40.0 | 66.2 | 55.3 | 45.6 | **34.0** | **100.0** | 70.1 | 61.8 | 59.5 |
| | | SAA | 73.5 | 58.9 | 45.7 | 79.2 | 63.4 | 50.7 | 32.8 | 100.0 | 75.8 | 71.4 | 65.8 |

that SAA substantially narrows the performance gap between black-box and white-box attacks, thereby facilitating a more comprehensive evaluation of the adversarial robustness of existing models.

**Performance comparison with attacks on ViTs.** To further explore the cross-architecture transferability, we evaluate adversarial attacks on ViTs, including SGM [59], PatchOut [58], PNA [58], and TGR [66]. Here, we choose the surrogate model as ViT-B and the witness model as Res50. In PatchOut, SAA first improves the transferability between ViTs, for example, the ASR from ViT-B to Swin-B is improved by 42.6%. Secondly, SAA greatly improves the transferability from ViT to CNNs, for example, the ASR is improved by 45.0% and 42.0% on Res50 and DN121 respectively. On SGM, PNA, and TGR, SAA also achieves stronger cross-architecture transferability without modifying the forward and backward propagation of the model.

**Summary.** Although we conduct 16 combinations and evaluate on 6 CNNs and 4 ViTs, showing that SAA's improvement is not limited to specific choices of witness models. Based on the above experiments, we further summarize the empirical guidance for the selection of surrogate and witness models: **i)** Self-alignment (surrogate and witness models are consistent) usually only provides minor improvements (it is difficult to learn unique structural features), and alignment between different models improves more significantly. **ii)** When the surrogate model is a ViT-like model and the witness model is a CNN-like model, better cross-architecture transferability can be obtained. This is mainly because ViT has stronger performance, while CNN further provides unique structural features. It should be noted that the target models are unknown in the transfer attack, so no information of the target models can be used for alignment.

Table 3: SAA has stronger adversarial transferability with existing transfer attacks.

| Attack | Target Model | | | | | | | | | | Avg. ASR (%) |
| --- | --- | --- | --- | --- | --- | --- | --- | --- | --- | --- | --- |
| | CNNs | | | | | | ViTs | | | | |
| | Res18 | Res50 | Res101 | VGG19 | DN121 | Inc-v3 | ViT-B | Swin-B | PVT-v2 | MobViT | |
| MI | 57.7 | **99.9** | 58.1 | 54.2 | 55.1 | 39.0 | 9.4 | 33.0 | 38.0 | 35.7 | 42.2 |
| MI-SAA | **84.1** | 99.6 | **74.7** | **80.3** | **81.8** | **65.7** | **24.3** | **48.7** | **52.9** | **62.5** | **63.9** |
| NI | 58.9 | **100.0** | 63.2 | 59.3 | 61.4 | 40.0 | 9.6 | 37.4 | 41.8 | 38.1 | 45.5 |
| NI-SAA | **86.1** | 99.9 | **76.3** | **82.2** | **83.7** | **67.6** | **24.0** | **50.6** | **55.7** | **64.8** | **65.7** |
| GI | 57.3 | **100.0** | 62.3 | 60.5 | 60.5 | 40.7 | 12.5 | 36.8 | 40.7 | 39.6 | 45.7 |
| GI-SAA | **86.5** | 99.7 | **78.8** | **83.9** | **84.8** | **70.4** | **27.6** | **52.7** | **55.8** | **66.3** | **67.4** |
| DI | 44.1 | **95.8** | 41.7 | 56.1 | 44.2 | 26.1 | 5.6 | 30.9 | 36.7 | 35.1 | 35.6 |
| DI-SAA | **74.6** | 94.4 | **61.1** | **81.1** | **73.2** | **53.1** | **10.6** | **44.0** | **50.8** | **63.6** | **56.9** |
| TI | 38.5 | **99.9** | **37.8** | 33.9 | 36.1 | 24.2 | 5.4 | 21.0 | 29.0 | 21.1 | 27.4 |
| TI-SAA | **59.7** | 94.9 | 35.2 | **50.6** | **54.5** | **40.6** | **9.5** | **22.8** | **29.3** | **33.6** | **37.3** |
| SSA | 75.8 | **99.9** | **78.6** | 76.0 | 77.8 | 57.0 | 16.5 | **48.5** | 55.0 | 50.5 | 59.5 |
| SSA-SAA | **91.5** | 99.5 | 77.8 | **85.7** | **88.4** | **74.9** | **23.4** | 46.7 | **57.1** | **66.0** | **67.9** |
| DI-MI | 65.5 | 97.0 | 65.0 | 74.7 | 65.7 | 49.1 | 16.4 | 49.0 | 54.9 | 57.9 | 55.4 |
| DI-MI-SAA | **91.9** | **98.7** | **84.9** | **94.1** | **90.5** | **78.3** | **34.1** | **69.8** | **76.1** | **86.8** | **78.5** |
| TI-MI | 61.4 | **99.9** | 60.5 | 60.9 | 60.9 | 44.3 | 15.2 | 37.4 | 42.3 | 41.8 | 47.2 |
| TI-MI-SAA | **84.8** | 99.3 | **71.9** | **79.0** | **81.8** | **69.1** | **27.0** | **45.0** | **52.8** | **62.4** | **63.8** |
| SSA-MI | 89.6 | **99.9** | 92.2 | 89.5 | 91.0 | 77.6 | 39.2 | **74.4** | 76.4 | 76.3 | 78.5 |
| SSA-MI-SAA | **96.3** | 99.8 | **95.6** | **96.5** | **97.2** | **91.5** | **46.3** | 74.1 | **80.4** | **88.4** | **85.1** |
| SSA-DI-TI-MI | 93.5 | 98.5 | 92.3 | 95.0 | 93.7 | 85.5 | **55.9** | **83.2** | 87.1 | 89.8 | 86.2 |
| SSA-DI-TI-MI-SAA | **97.5** | **98.8** | **93.8** | **97.6** | **96.7** | **94.3** | 53.6 | 81.6 | **84.4** | **94.7** | **88.2** |

Table 4: SAA further improves the adversarial transferability of adversarial attacks on ViTs.

| Attack | Target Model | | | | | | | | | | Avg. ASR (%) |
| --- | --- | --- | --- | --- | --- | --- | --- | --- | --- | --- | --- |
| | CNNs | | | | | | ViTs | | | | |
| | Res18 | Res50 | Res101 | VGG19 | DN121 | Inc-v3 | ViT-B | Swin-B | PVT-v2 | MobViT | |
| SGM | 82.9 | 67.6 | 59.4 | 81.2 | 75.4 | 71.3 | **99.7** | 83.3 | 72.7 | 78.8 | 78.3 |
| SGM-SAA | **91.1** | **79.8** | **73.3** | **87.5** | **87.3** | **80.9** | 99.5 | **90.5** | **82.6** | **86.3** | **86.6** |
| PatchOut | 45.6 | 27.4 | 20.3 | 45.5 | 36.1 | 33.9 | 93.0 | 41.0 | 34.2 | 40.5 | 43.3 |
| PatchOut-SAA | **76.5** | **72.4** | **70.3** | **79.4** | **78.1** | **71.3** | **94.7** | **83.6** | **77.2** | **76.8** | **78.7** |
| PNA | 61.2 | 45.0 | 38.1 | 60.8 | 54.8 | 49.0 | **99.6** | 66.3 | 55.8 | 56.8 | 60.3 |
| PNA-SAA | **82.7** | **78.1** | **73.4** | **85.6** | **84.0** | **75.1** | 97.4 | **89.3** | **80.3** | **82.1** | **83.3** |
| TGR | 74.0 | 55.6 | 48.4 | 73.2 | 66.6 | 59.0 | **99.3** | 74.5 | 61.6 | 69.6 | 69.6 |
| TGR-SAA | **85.9** | **78.1** | **71.5** | **87.4** | **85.6** | **79.6** | **99.3** | **89.1** | **81.0** | **86.2** | **85.1** |

## 3.3 Ensemble Attacks

The ensemble of multiple surrogate models can further improve adversarial transferability [13], so we want to explore whether SAA can further improve transferability. In the setting of ensemble attacks, we follow the setting of MI [13] and use the logits of multiple surrogate models for integration. Here we choose multiple groups of settings, such as Res50 (ViT-B) means that the surrogate model is Res50 and the witness model is ViT-B. Res50+ViT-B means the logits of these two models are integrated, and so on. As shown in Table 5, SAA further improves the performance of ensemble attacks. First, the alignment-based method is not as good as the direct integration strategy in white-box performance, but it has significant improvement in black-box transferability. For example, on (Res50+ViT-B), MA and SAA are 9.1% and 20.8% higher than the direct ensemble ASR, respectively. Then, MA directly constrains global features and does not take into account the huge differences in features between different structures, which may lead to degradation of transferability, such as on Swin-B. Finally, SAA has achieved state-of-the-art transferability and further enhance ensemble attacks because it considers spatial and adversarial features from a structural perspective.

## 3.4 Adversarial Defenses

To illustrate the effect of SAA in the face of adversarial defenses, we choose Inc-v3$_{ens3}$ as the target model and evaluate its performance on 12 adversarial defenses, including HGD [28], R&P [60],

Table 5: SAA improves the adversarial transferability of ensemble attacks.

| Surrogate | Attack | CNNs | | | | | | ViTs | | | | Avg. ASR (%) |
|---|---|---|---|---|---|---|---|---|---|---|---|---|
| | | Res18 | Res50 | Res101 | VGG19 | DN121 | Inc-v3 | ViT-B | Swin-B | PVT-v2 | MobViT | |
| Res50 | n/a | 57.7 | 99.9 | 58.1 | 54.2 | 55.1 | 39.0 | 9.4 | 33.0 | 38.0 | 35.7 | 42.2 |
| ViT-B | n/a | 48.6 | 41.0 | 31.9 | 58.1 | 50.8 | 44.4 | 100.0 | 58.1 | 44.7 | 47.4 | 47.2 |
| Res50+ViT-B | n/a | 61.3 | 100.0 | 63.0 | 60.8 | 61.2 | 45.0 | 97.6 | 50.4 | 51.3 | 48.1 | 55.1 |
| Res50(Res50)+Res50(ViT-B) | MA | 79.3 | 100.0 | 74.5 | 72.3 | 80.0 | 56.3 | 20.3 | 47.4 | 51.7 | 52.3 | 64.2 |
| | SAA | 86.8 | 100.0 | 85.9 | 83.9 | 86.7 | 70.0 | 30.7 | 60.4 | 64.1 | 69.6 | 75.9 |
| Res50+DN121+ViT-B+Swin-B | n/a | 65.4 | 100.0 | 78.0 | 79.1 | 99.8 | 62.0 | 93.3 | 100.0 | 77.4 | 68.9 | 71.8 |
| Res50(Res50)+Res50(DN121)+ | MA | 87.0 | 99.8 | 85.4 | 83.2 | 90.0 | 67.2 | 28.0 | 60.3 | 66.2 | 63.9 | 75.5 |
| Res50(ViT-B)+Res50(Swin-B) | SAA | 93.5 | 100.0 | 92.9 | 92.1 | 95.1 | 82.3 | 39.6 | 71.8 | 80.4 | 81.7 | 87.2 |

Table 6: SAA improves adversarial transferability against adversarial defenses.

| Attack | HGD | R&P | NIPS-r3 | JPEG | FD | RS | Bit-Red | NRP | Diffpure | Inc-v3$_{ens3}$ | Inc-v3$_{ens4}$ | IncRes-v2$_{ens}$ | Avg. ASR (%) |
|---|---|---|---|---|---|---|---|---|---|---|---|---|---|
| MI | 42.5 | 21.9 | 25.3 | 33.9 | 42.4 | 23.6 | 29.3 | 6.7 | 13.8 | 33.3 | 31.3 | 23.3 | 27.3 |
| MI-SAA | 73.3 | 57.8 | 60.4 | 69.9 | 65.6 | 39.6 | 42.3 | 12.0 | 22.4 | 68.9 | 65.5 | 57.9 | 53.0 |
| SSA-DI-TI-MI | 93.7 | 89.6 | 90.2 | 91.9 | 89.7 | 82.3 | 81.7 | 14.8 | 71.1 | 92.5 | 91.1 | 89.8 | 81.5 |
| SSA-DI-TI-MI-SAA | 96.0 | 93.2 | 94.8 | 95.1 | 94.0 | 89.8 | 85.5 | 20.2 | 78.8 | 95.7 | 94.5 | 93.3 | 85.9 |

Table 7: Ablation study on alignment modules.

| Module | | | Target Model | | | | | | | | | | Avg. ASR (%) |
|---|---|---|---|---|---|---|---|---|---|---|---|---|---|
| Spatial | | Adversarial | CNNs | | | | | | ViTs | | | | |
| Global | Local | | Res18 | Res50 | Res101 | VGG19 | DN121 | Inc-v3 | ViT-B | Swin-B | PVT-v2 | MobViT | |
| | | | 57.7 | 99.9 | 58.1 | 54.2 | 55.1 | 39.0 | 9.4 | 33.0 | 38.0 | 35.7 | 42.2 |
| ✓ | | | 60.4 | 99.8 | 56.4 | 60.3 | 67.3 | 44.3 | 12.1 | 35.6 | 37.2 | 39.0 | 45.8 |
| ✓ | ✓ | | 67.5 | 98.0 | 58.1 | 69.1 | 70.1 | 49.9 | 12.2 | 37.7 | 38.9 | 47.7 | 50.1 |
| ✓ | | ✓ | 81.6 | 97.9 | 68.1 | 53.0 | 79.8 | 63.6 | 25.5 | 47.9 | 48.7 | 57.3 | 58.4 |
| ✓ | ✓ | ✓ | 84.1 | 99.6 | 74.7 | 80.3 | 81.8 | 65.7 | 24.3 | 48.7 | 52.9 | 62.5 | 63.9 |

NIPS-r3[4], JPEG [19], FD [36], RS [10], Bit-Red [62], NRP [41], Diffpure [43], Inc-v3$_{ens3}$ [50], Inc-v3$_{ens4}$ [50], and IncRes-v2$_{ens}$ [50]. Here we use the ensemble attack setting, and the surrogate models are Res50(Res50)+Res50(DN121)+Res50(ViT-B)+Res50(Swin-B). As shown in Table 6, although adversarial defenses weaken transferability, SAA still achieves a significant improvement in adversarial transferability compared to the origin surrogate models.

## 3.5 Ablation Studies

We select ResNet-50 as the surrogate model and ViT-B as the witness model for ablation studies (see **Appendix** C for more details, including training epochs, distance metric, GPU memory and computing cost, self-adversarial strategy).

**Alignment Module.** In spatial-aware alignment, 'global' represents $\mathcal{L}_{global}$ (Equation 1), while 'local' represents $\mathcal{L}_{local}$ (Equation 2). Similarly, 'adversarial' represents $\mathcal{L}_{AA}$ (Eqution 5) of adversarial-aware alignment. As shown in Table 7, when global features are introduced into the alignment, the transferability of the aligned surrogate model will increase by 3.6%. Based on $\mathcal{L}_{global}$, when only local features are introduced, the overall transferability is improved by 4.3% due to better alignment of features of different architectures in local regions, especially by 8.7% on MobViT. Since intermediate layer features between models are diverse and different, we choose last layer features that summarize the spatial information, which is a simple and effective strategy. Based on $\mathcal{L}_{global}$, when only adversarial features are introduced, the transferability is greatly improved, reaching 12.6% ASR, and the improvement is significant on ViTs. Finally, by integrating all features, the aligned surrogate model achieves state-of-the-art transferability.

## 4 Conclusions

In this study, we introduce a novel technique called Spatial Adversarial Alignment (SAA), which incorporates an alignment loss function and utilizes a witness model to fine-tune a surrogate model by focusing on both spatial-aware and adversarial-aware alignments. Through experimental analysis, we

---

[4] https://github.com/anlthms/nips-2017/tree/master/mmd

show that leveraging these spatial and adversarial features for model alignment significantly enhances the adversarial transferability of surrogate models, with a particularly pronounced improvement in their cross-architecture capabilities. SAA not only integrates seamlessly with existing transfer attack strategies but also further amplifies adversarial transferability, thereby contributing to a more complete evaluation of the adversarial robustness of DNNs.

**Boarder Impacts.** Adversarial examples by SAA exhibit enhanced adversarial transferability, especially in cross-architecture capabilities. This poses a huge threat to the deployment of real-world applications. Simultaneously, it is also conducive to better evaluating their adversarial robustness.

**Limitations.** The huge gaps between network architectures limit transferability. While SAA aligns unique features under both spatial and adversarial conditions to mitigate these gaps, it does not fully resolve them. Then, SAA also introduces some computing costs during the alignment but no extra costs during attacking. In addition, a theoretical analysis is lacking. We have made significant progress, but there is still much to be done in addressing this issue.

**Acknowledgments.** This work was supported by National Natural Science Foundation of China (No.62576109, 62072112).

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

- We provide the implementation of our Spatial Adversarial Alignment (SAA) in the **Core Codes** of **Supplementary Material**.

# A  Related Work

## A.1  Transfer attacks on CNNs

Early transfer attacks are mainly conducted between CNNs, and the most popular methods were advanced optimization [13, 29, 51], data augmentation [61, 14, 37], and model modification [59, 21, 58].

**Advanced Optimization.** [29] compare adversarial attacks to model training: better optimization methods can obtain models with better generalization, and therefore also generate adversarial examples with higher transferability. FGSM [18] is the earliest gradient-based transfer attack, which was then extended to I-FGSM [27]. The subsequent advance optimization further improves the transferability by introducing momentum [13, 29, 53, 51] and smoothness [44].

**Data Augmentation.** Data augmentation serves as an effective strategy to prevent model overfitting, achieving state-of-the-art performance in model generalization [65, 12]. Building on this principle, numerous adversarial attacks incorporate various transformations to enhance adversarial transferability, including modifications in size [61], scale [29], mixup [54], and frequency domain [37] adjustments. This integration aims to mitigate the overfitting of adversarial examples to the surrogate model, thereby increasing their effectiveness across different models.

**Model Modification.** According to certain characteristics of the model, modifying the parameters of the surrogate model or changing the forward or backward propagation can also improve the transferability. Skip Gradient Method (SGM) [59] using more gradients from the skip connections rather than the residual modules, allows one to craft adversarial examples with high transferability. Similarly, Linear Backpropagation (LinBP) [21] and Backward Propagation Attack (BPA) [55] concentrate on non-linear activations by modifying the ReLU derivatives to enhance attack transferability. Model Alignment (MA) [38] promotes alignment of model predictions through an alignment loss relative to a witness model, with the aim of capturing shared features across models. However, MA overlooks spatial and adversarial feature alignment across architectures, limiting its effectiveness. Unlike these methods, our SAA requires no modifications to the forward or backpropagation processes, enabling the efficient generation of highly transferable adversarial examples with minimal training overhead. In contrast, LinBP and BPA, involve altering backpropagation or even full model retraining, incurring significantly higher computational costs.

## A.2  Transfer attacks on ViTs

Current transfer attacks for ViTs largely adapt methods developed for CNNs. Pay No Attention (PNA) [58] method extends Skip Gradient Method (SGM) to ViTs by omitting the gradient computation of attention blocks during back-propagation, thereby enhancing adversarial transferability. PatchOut [58] strategy selects a random subset of image patches to compute the gradient at each iteration, functioning as an image transformation technique to increase transferability. Then, Self-Ensemble (SE) [42] approach employs the class token at each layer with a shared classification

head to create an ensemble model, facilitating optimized perturbation; however, many ViTs, such as Visformer [8] and CaiT [49], lack sufficient class tokens to build this ensemble. Additionally, Token Refinement (TR) [42] module fine-tunes class tokens to further boost transferability. Recently, Token Gradient Regularization (TGR) [66] works from the perspective of variance reduction, stabilizing the gradient direction to prevent adversarial examples from getting stuck in poor local optima. Distinct from these approaches, SAA is the first method to specifically analyze architectural differences across models. By leveraging shared features between different architectures, SAA enables the creation of more generalized surrogate models that integrate seamlessly with optimization and data augmentation methods, ultimately achieving state-of-the-art transferability.

---

**Algorithm 1** Loss $\mathcal{L}_{SAA}$

---

**Require:** Input $x$, surrogate model $f_{\theta_s}$, witness model $f_{\theta_w}$
**Ensure:** $\mathcal{L}_{SAA}$
 1: **#Preliminaries**
 2: **if** $f_\theta$ is CNN **then**
 3:      Compute global logits:
             $f_\theta(x) = \text{Classifier}(\text{GlobalAvgPooling}(Backbone(x)))$
 4:      Obtain local regions collection $z$:
             $z_\theta(x) = \text{Classifier}(Backbone(x))$
 5: **else if** $f$ is ViT **then**
 6:      $z' = \text{TransformerEncoder}(x)$
 7:      Compute global logits:
             $f_\theta(x) = \text{Classifier}(z'[0])$
 8:      Obtain local regions collection $z$:
             $z_\theta(x) = \text{Classifier}(z'[1 :])$
 9: **end if**
10: **#Step 1: Compute spatial-aware alignment loss**
11: Obtain global logits $f_{\theta_s}(x)$ and $f_{\theta_w}(x)$
12: Compute $\mathcal{L}_{global}(x; \theta_s) = D_{\text{KL}}(f_{\theta_s}(x), f_{\theta_w}(x))$
13: **for** each local region $q$ **do**
14:      Obtain local logits $z_{\theta_s}^{[q]}$ and $z_{\theta_w}^{[q]}$
15:      Derive pseudo-label $\hat{y}_{\theta_w}^{[q]} = \arg\max(z_{\theta_w}^{[q]}(x))$
16:      Compute local alignment loss for region $q$: $D_{\text{CE}}(z_{\theta_s}^{[q]}(x), \hat{y}_{\theta_w}^{[q]})$
17: **end for**
18: Calculate      total      local      alignment      loss:      $\mathcal{L}_{local}(x; \theta_s)$      $=$
      $\frac{1}{HW}\sum_{q=1}^{HW} D_{\text{CE}}(z_{\theta_s}^{[q]}(x), \arg\max(z_{\theta_w}^{[q]}(x)))$
19: Calculate spatial-aware alignment loss: $\mathcal{L}_{SA}(x; \theta_s) = \mathcal{L}_{global}(x; \theta_s) + \gamma \cdot \mathcal{L}_{local}(x; \theta_s)$
20: **#Step 2: Compute adversarial-aware alignment loss**
21: Initialize adversarial example $x_{\text{adv}}^{(0)} = x$
22: **for** each iteration $t$ **do**
23:      $x_{adv}^{(t+1)} = \Pi_{x,\epsilon}\left(x_{\text{adv}}^{(t)} + \alpha \cdot \text{sign}\left(\nabla_x D_{\text{KL}}\left(f_{\theta_s}(x_{\text{adv}}^{(t)}), f_{\theta_w}(x)\right)\right)\right)$
24: **end for**
25: Obtain final adversarial example $x_{\text{adv}} = x_{\text{adv}}^{(T)}$
26: Calculate  adversarial-aware  alignment  loss:  $\mathcal{L}_{\mathcal{AA}}(x_{adv}; \theta_s)$  $=$  $\mathcal{L}_{global}(x_{adv}; \theta_s) + \omega \cdot$
      $\mathcal{L}_{local}(x_{adv}; \theta_s)$
27: **#Step 3: Calculate spatial adversarial alignment loss**
28: $\mathcal{L}_{SAA}(x; \theta_s) = \mathcal{L}_{SA}(x; \theta_s) + \kappa \cdot \mathcal{L}_{AA}(x_{adv}; \theta_s)$
29: **Return** $\mathcal{L}_{SAA}$

---

# B   Loss Calculation

We introduce the calculation of loss in detail, as shown in Algorithm 1. For the specific implementation, please refer to the code provided in **Supplementary Material**.

Table 8: Ablation study on distance metrics of global features.

| Loss | Target Model | | | | | | | | | | Avg. ASR (%) |
| | CNNs | | | | | | ViTs | | | | |
| | Res18 | Res50 | Res101 | VGG19 | DN121 | Inc-v3 | ViT-B | Swin-B | PVT-v2 | MobViT | |
|---|---|---|---|---|---|---|---|---|---|---|---|
| n/a | 57.7 | 99.9 | 58.1 | 54.2 | 55.1 | 39.0 | 9.4 | 33.0 | 38.0 | 35.7 | 42.2 |
| KL | 84.1 | 99.6 | 74.7 | 80.3 | 81.8 | 65.7 | 24.3 | 48.7 | 52.9 | 62.5 | 63.9 |
| TV | 62.1 | 97.8 | 56.2 | 67.5 | 63.9 | 47.6 | 9.4 | 32.2 | 37.2 | 44.1 | 46.7 |
| JS | 83.3 | 99.1 | 70.0 | 78.2 | 80.9 | 63.5 | 22.8 | 46.4 | 49.3 | 58.5 | 61.4 |
| NCE | 78.8 | 99.3 | 69.1 | 77.0 | 76.6 | 60.3 | 21.7 | 46.2 | 48.2 | 55.8 | 59.3 |

Table 9: Ablation study on different supervision in self-adversarial strategy.

| Source | Witness | Attack | Res18 | Res50 | Res101 | VGG19 | DN121 | IncV3 | ViT-B | Swin-B | PVT-v2 | MobViT | Avg. ASR (%) |
|---|---|---|---|---|---|---|---|---|---|---|---|---|---|
| | | MA | 74.2 | 99.2 | 63.5 | 69.3 | 72.8 | 51.5 | 18.5 | 41.5 | 42.7 | 47.4 | 53.5 |
| Res50 | ViT-B | SAA $(f_{\theta_w}(x))$ | **84.1** | **99.6** | **74.7** | **80.3** | **81.8** | **65.7** | **24.3** | **48.7** | **52.9** | **62.5** | **63.9** |
| | | SAA $(f_{\theta_s}(x))$ | 81.1 | 96.6 | 68.7 | 77.5 | 79.5 | 60.6 | 19.4 | 45.9 | 46.2 | 57.6 | 59.6 |

## C   Ablation Study

We select ResNet, DenseNet, and Swin, all of which have won Best Paper Awards, and ViT, which is a pioneering work. These models are representative of both CNNs and ViTs and their weights are all from *timm*, ensuring the generalization of our conclusions. We conduct 16 combinations of experiments and evaluate on 6 CNNs and 4 ViTs, showing that SAA's improvement is not limited to specific choices of witness models.

**Training Epochs.** In Section 3.2, we reveal the powerful potential of SAA for adversarial transferability after training for only one epoch. Furthermore, we explore the performance difference after training for multiple epochs, as shown in Figure 3. We calculate the average attack success rate except for the Res50 surrogate model itself and find that with the increase of epochs, the adversarial transferability of the aligned surrogate model is further improved, reaching convergence around the 9-th epoch. Compared with MA, SAA can achieve higher transferability in small epochs, and after multiple rounds of training, the transferability has a higher upper limit, which shows the importance of using spatial and adversarial features for model alignment.

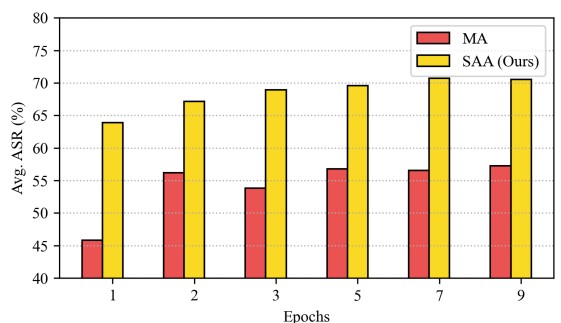

Figure 3: Ablation study on training epochs.

**Distance Metric**. There are many ways to align the output of global features. To better align the global features, we select Kullback–Leibler Divergence (KL), Total Variation (TV), Jensen-Shannon Divergence (JS), and NCE [2] loss for evaluation. We select ResNet-50 as the surrogate model and ViT-B as the witness model for ablation studies. As shown in Table 8, KL exceeds the next best JS by 2.5%, so we choose KL as the metric.

**GPU Memory and Computing Cost.** SAA's GPU memory is split into alignment and attack. In alignment, we infer the witness model and train the surrogate model for one epoch, so the memory usage is the sum of both. When the surrogate is Res50 and the witness is ViT-B, SAA's GPU memory is only 2794MB with a batch size of 1. Once alignment is complete, attack costs only are an aligned surrogate model, making SAA's memory the same as SOTA attacks without extra memory. Thus, its GPU memory is acceptable. For computing cost, SAA needs about 10 hours' training time on ImageNet with batch size of 64 under Nvidia RTX 3090. The total computational cost of MA is 2 hours with the same setting. Since adversarial-aware alignment introduces the solution of adversarial examples, in the practice of SAA, model training takes about 2 hours, and solving adversarial examples takes about 8 hours, for a total of 10 hours. Note that the main computational bottleneck is actually obtaining adversarial examples, not model training. Some newer adversarial

Table 10: Ablation study on label supervision in local alignment.

| Source | Witness | Attack | Res18 | Res50 | Res101 | VGG19 | DN121 | IncV3 | ViT-B | Swin-B | PVT_v2 | MobViT | Avg. ASR (%) |
|--------|---------|--------|-------|-------|--------|-------|-------|-------|-------|--------|--------|--------|--------------|
| Res50 | ViT-B | SAA-MI (Ours) | **84.1** | **99.6** | **74.7** | **80.3** | **81.8** | **65.7** | **24.3** | **48.7** | **52.9** | **62.5** | **63.9** |
| | | SAA-MI (soft) | 81.9 | 96.4 | 65.9 | 74.5 | 77.7 | 60.8 | 18.8 | 44.4 | 43.1 | 53.0 | 57.8 |

Table 11: SAA improves the adversarial transferability of feature-based transfer attacks.

| Attack | Target Model | | | | | | | | | | Avg. ASR (%) |
|--------|------|-------|--------|-------|-------|-------|-------|--------|--------|--------|--------------|
| | CNNs | | | | | | ViTs | | | | |
| | Res18 | Res50 | Res101 | VGG19 | DN121 | Inc-v3 | ViT-B | Swin-B | PVT-v2 | MobViT | |
| FDA | 55.2 | 31.6 | 36.8 | 52.0 | 43.6 | 27.1 | 2.7 | 21.0 | 24.5 | 27.8 | 32.3 |
| FDA-SAA | **82.9** | **79.0** | **72.5** | **79.1** | **77.1** | **60.1** | **14.4** | **39.5** | **43.4** | **58.4** | **58.6** |
| FDA-MI | 75.0 | 46.1 | 60.6 | 74.0 | 65.3 | 46.0 | 8.4 | 27.3 | 29.4 | 47.0 | 48.1 |
| FDA-MI-SAA | **89.7** | **88.2** | **81.7** | **86.9** | **85.3** | **73.1** | **21.9** | **41.7** | **44.7** | **68.9** | **66.0** |

attack techniques [57] may be able to accelerate this process and reduce the time to 1/3 of the original, which is an auspicious research direction.

**Differnet supervision in self-adversarial strategy.** In SAA, adversarial-aware alignment is performed using the self-adversarial strategy, where the supervision comes from global features of the witness model. Here, we investigate whether supervision from global features of the surrogate model or global features of the witness model is more effective, and compare both approaches with the MA [38] baseline. The results in Table 9 show that SAA ($f_{\theta_w}(x)$) outperforms SAA ($f_{\theta_s}(x)$) by 4.3% in ASR, indicating that $f_{\theta_w}(x)$ provides stronger supervision for adversarial-aware alignment, leading to highly transferable adversarial examples. This highlights the effectiveness of alignment using $f_{\theta_w}(x)$ and its superiority over $f_{\theta_s}(x)$, demonstrating that leveraging witness model features enhances adversarial robustness.

**Label supervision in local alignment.** We add experiments using soft labels (logits) and the KL divergence loss function for local alignment, as shown in Table 10. The surrogate model is Res50 and the witness model is ViT-B. In this scenario, using hard labels (local pseudo-labels) achieves higher adversarial transferability (improving ASR by 6.1%).

# D   Feature-based Transfer Attacks

Here we combine SAA with feature-based transfer attacks for experiments, as shown in Table 11. Here we select ResNet-50 as the surrogate model and ViT-B as the witness model. Since the aligned surrogate model after SAA can learn more common features, SAA further improves the attack performance of FDA [17]. On FDA and FDA-MI, SAA improves the black-box attack success rate by 25.7% and 17.1%, respectively.

# E   Grad-CAM Visualization

We present Grad-CAM visualizations on two target models: ResNet-101 and ViT-S, as shown in Figure 4. Each set consists of three columns: the first column shows the Grad-CAM heatmap of the target model on clean input; the second column displays the heatmap for adversarial examples generated by the unaligned model; the third column corresponds to adversarial examples generated by SAA aligned model. The visualizations indicate that adversarial examples from the unaligned model produce heatmap distributions more similar to those of clean inputs, whereas adversarial examples from the aligned model effectively disrupt the target model's attention distribution, leading to a more thorough attack.

# F   Security Implications

**Potential misuse.** SAA improves the cross-architecture adversarial transferability of surrogate models, which means attacks against unknown systems pose a higher threat. For example, for a face forgery detection system [3], open-source models are typically CNNs, while the most advanced

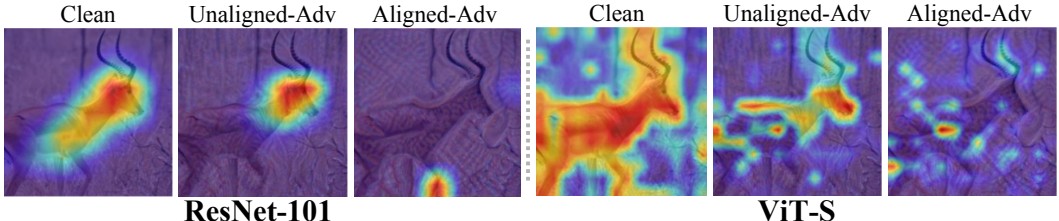

Figure 4: Grad-CAM on target models.

commercial applications may already be ViT-based models. In this case, using existing CNN detectors as surrogate models, combined with SAA, can lead to successful attack on commercial applications.

**Mitigation strategies.** We think that potential defense strategies primarily involve introducing adversarial training and increasing model diversity. The former recommends introducing a small number of adversarial examples during model training to improve robustness against adversarial examples. The latter recommends introducing multi-model ensemble training during training to increase the model's feature complexity and enhance deployment robustness.

