# OpenReview forum: "Boosting Adversarial Transferability with Spatial Adversarial Alignment"
_NeurIPS.cc/2025/Conference — NeurIPS 2025 poster_

### Official Review · Reviewer_hgUT · 2025-06-30

**Clarity:** 4
**Significance:** 3
**Originality:** 3
**Rating:** 5
**Confidence:** 5

**Summary:**

This paper explores the transferability of adversarial examples across architectures. The core insight is that previous work doesn't perturb the common features between different models. Therefore, the authors propose a technique termed Spatial Adversarial Alignment (SAA), which employs an alignment loss and leverages a witness model to fine-tune the surrogate model. Through spatial-aware and adversarial-aware alignment, the surrogate model is trained to concentrate on the common features extracted by the witness model. This facilitates adversarial attacks on these shared features, thereby yielding perturbations that exhibit enhanced transferability. Experiments on 6 CNNs and 4 ViTs show that SAA has state-of-the-art adversarial transferability, especially in cross-model transferability. In addition, the authors also provide empirical guidance for model selection and release code in the supplementary material.

**Questions:**

See Weaknesses.

**Ethical Concerns:**

["NO or VERY MINOR ethics concerns only"]

**Final Justification:**

The review meets my requirements. I recommend: 5: Accept!

**Limitations:**

Yes, this paper has chaimed its limitations and I think it is enough.

**Quality:**

4

**Strengths And Weaknesses:**

Strengths:

This paper is of high value because exploring the transferability of adversarial examples across architectures has important security implications for the deployment of practical models. The authors further summarize the empirical guidance for the model selection in SAA, which is beneficial for practice.
The proposed aligned-based method is novel, focusing on the common features in space and adversarial space that were ignored in previous work, and can be seamlessly integrated with existing optimization and data augmentation-based methods to achieve stronger performance.
Extensive experiments in the main body and appendix verify the effectiveness of SAA. The code released in the supplementary material provides reproducibility.
Weaknesses:

There is a lack of description on the difference between alignment and direct ensemble model attack. From the perspective of model configuration, the two are very similar.
Why does the self-aligned strategy also achieve certain improvements?
The differences between SAA and other attacks on ViTs such as SGM, PNA, and TGR were not discussed.

---

> ### Author Rebuttal · Authors · 2025-07-31
>
> Thank you for your recognition and support of our work. We hope you can be our champion in future discussions.
>
> ---
>
> **Weakness 1:** Difference between alignment and direct ensemble model attack.
>
> We sincerely appreciate the reviewer's insightful question. To clarify the difference between alignment-based methods and direct ensemble model attacks: While both aim to enhance adversarial transferability, their approaches are distinct. Alignment-based methods, such as ours, focus on **improving surrogate models themselves** by leveraging witness models to align structural features, thus generating more transferable adversarial examples. complementary strengths. These two strategies are complementary rather than conflicting.
>
> Notably, our alignment-based method (SAA) can **further enhance ensemble attacks**, as demonstrated in Table 5 of the main body. For example, when combined with an ensemble of Res50+ViT-B, SAA improves the attack success rate (ASR) by **20.8%** compared to the direct ensemble baseline, outperforming even the state-of-the-art MA method (+9.1%). This confirms that Alignment-based refinement of surrogate models provides orthogonal benefits to ensemble strategies, strengthening their transferability in black-box settings. We hope this clarifies their distinct contributions.
>
> ---
>
> **Weakness 2:** The improvement of the self-aligned strategy.
>
> While the unique spatial features in the self-aligned strategy are relatively limited, the surrogate models can still learn features related to adversarial patterns and category relationships. In terms of adversarial features, the adversarial-aware alignment enables dynamic learning of adversarial characteristics. Since the witness model itself is frozen, the surrogate models can align with adversarial features without deviating significantly from their own inherent properties.
> Additionally, the surrogate models can still acquire high confidence in the correct categories under soft labels, which also includes the "ambiguity" towards similar categories. Such soft labels guide the surrogate models to learn more robust feature representations, thereby enhancing the transferability of the generated adversarial examples.
> We believe these factors collectively contribute to the improvements brought by the self-aligned strategy. Thank you again for your valuable feedback.
>
> ---
>
>
> **Weakness 3:** The differences between SAA and other attacks on ViTs such as SGM, PNA, and TGR.
>
> We appreciate your attention to the differences between SAA and other attacks on ViTs, such as SGM, PNA, and TGR. As shown in Table 4 of our paper, SAA demonstrates stronger cross-architecture transferability compared to these methods, which preliminarily reflects the superiority of SAA in common settings.​
>
> Furthermore, we have provided a detailed discussion on SGM, PNA, and TGR in Appendix A.2. Specifically, the Pay No Attention (PNA) method extends the Skip Gradient Method (SGM) to ViTs by omitting the gradient computation of attention blocks during back-propagation, thereby enhancing adversarial transferability. Token Gradient Regularization (TGR) operates from the perspective of variance reduction, stabilizing the gradient direction to prevent adversarial examples from getting stuck in poor local optima.​
>
> Distinct from these methods, SAA is the first method to specifically analyze architectural differences across models. It enables the creation of more generalized surrogate models that integrate seamlessly with optimization and data augmentation methods without modifying the forward and backward propagation of the model. This unique design is what ultimately allows SAA to achieve state-of-the-art transferability.

---

> > ### Comment · Reviewer_hgUT · 2025-08-02
> > **Boosting Adversarial Transferability with Spatial Adversarial Alignment**
> >
> > ​​Thank you for the detailed responses, and your replies have fully addressed all the concerns I raised. Furthermore, I note that there is a clear consensus among all reviewers regarding the novelty and the solid experimental results of this paper. It is also valuable that the code has been released in the supplementary materials. I believe this work presents meaningful contributions that will inspire further discussion and advancement in the field of adversarial transferability. Accordingly, I recommend acceptance with a rating of "5: Accept".​

---

> > > ### Author Response · Authors · 2025-08-02
> > >
> > > Thanks for your recognition and support of our work. We will update the rebuttal into the final version. We hope this work can bring new inspiration to transfer attacks.

---

### Official Review · Reviewer_Qa5A · 2025-06-30

**Clarity:** 3
**Significance:** 2
**Originality:** 2
**Rating:** 4
**Confidence:** 2

**Summary:**

The authors tackle the problem of cross-architecture adversarial transferability of adversarial attacks. As different architectures have significantly different structures, existing methods are suboptimal when transfering from different model families (e.g., CNN to ViT). Thus, they propose a technique termed Spatial Adversarial Alignment (SAA) consisted of spatial-aware alignment and adversarial-aware alignment. SAA aims to adjust the surrogate model to extract features closely aligned with those of the witness model by aligning both the local and global features between a surrogate and a witness model. SSA performs such alignment both in the clean and adversarial setting. SSA demonstrates significant improvements compared to MA across a wide range of models and demonstrate improvements in transfer attacks and ensemble attacks.

**Questions:**

- What can we learn from the fact that some architectures of ViT transfers better to some CNN architectures? e.g., Swin-B to VGG?

**Ethical Concerns:**

["NO or VERY MINOR ethics concerns only"]

**Final Justification:**

Based on the authors' rebuttal, I maintain my positive score

**Limitations:**

Yes

**Quality:**

3

**Strengths And Weaknesses:**

**Strengths**
- Elegant yet practical idea – SAA keeps the training recipe minimal while still delivering substantial transfer-rate gains; the large, multi-model benchmark makes the effect hard to dismiss.
- Clear exposition – the narrative flows logically from motivation through method to experiments, making the paper easy to read even for newcomers.
- Comprehensive empirical evidence – the authors supply full ablations (loss terms, distance metrics, training length) and report results on six CNNs, four ViTs, ten attack variants, and a dozen defences, lending strong credibility to the claims.


**Weaknesses**
- Narrow comparative baseline – Table 2 pits SAA only against Model-Alignment (MA), an older 2022. Comparison to more modern methods is missing.

- Limited insight in section 2.3 – the section merely restates that the loss drives global-feature similarity and then confirms that the learned features are, in fact, similar.

- Weak qualitative evidence – the Grad-CAM comparison is based on just two images; furthermore, the surrogate heat-maps appear more diffused rather than genuinely mirroring the witness, so the visual argument feels anecdotal. A systematic, quantitative localisation metric would strengthen the case.

---

> ### Author Rebuttal · Authors · 2025-07-31
>
> **Weakness 1:** More modern methods is missing.
>
> In fact, we compared many transfer attacks from the past three years. Among aligned-based methods, **Model Alignment (MA) is quite new, having been accepted by ECCV 2024, not 2022.** The fact that SAA outperforms MA clearly demonstrates its state-of-the-art performance. Furthermore, experiments combining it with other transfer attacks also incorporate a lot of work from the past three years, such as GI (accepted by Expert Systems with Applications 2024).
>
> ---
>
> **Weakness 2 & Weakness 3:** Limited insight and Weak qualitative evidence in section 2.3.
>
> Thank you for your insightful comments, which help us clarify the focus and limitations of our analysis in Section 2.3. We appreciate the opportunity to address your concerns.
>
>
> First, we acknowledge that qualitatively and quantitatively dissecting differences between model features is inherently challenging. In Section 2.3, our core focus is to verify whether SAA improves the **spatial and adversarial features** of surrogate models after alignment, rather than analyzing local vs. global features in spatially-aware alignment. This distinction is important: the link between such feature improvements and practical transferability has already been validated through quantitative transferability analysis in Table 7, which demonstrates that enhanced feature alignment directly translates to stronger attack performance across architectures.
>
>
> Second, regarding the use of cosine similarity: since spatial features here refer to representations of entire images (not localized regions), cosine similarity serves as an intuitive and widely adopted metric to quantify global feature similarity between models. As shown in Table 1, two key observations emerge:
> 1. there is a significant inherent feature gap between diverse architectures (e.g., Res50 and Swin-B);
> 2. SAA consistently reduces this gap, with meaningful improvements (e.g., the 49.3% gain for Res50-Swin-B on adversarial examples). These results quantitatively confirm that SAA effectively identifies shared features between distinct models, which in turn enhances adversarial transferability.
>
>
> Finally, we recognize the limitation of the Grad-CAM examples provided in the main body, which are constrained by space. However, these visualizations do illustrate that aligned surrogates learn features more consistent with the witness model—for instance, in Figure 2, aligned surrogates on clean inputs exhibit feature distributions closer to the witness model’s patterns. To address the need for more evidence, we have included additional visualizations and analyses in Appendix E: these show that adversarial examples from unaligned surrogates retain heatmap distributions similar to clean inputs (indicating poor transfer), while those from aligned surrogates effectively disrupt the target model’s attention patterns, leading to more thorough attacks. This strengthens the case that SAA-driven alignment modifies features in a functionally meaningful way.
>
>
> We hope these clarifications address your concerns. Thank you again for pushing us to refine our analysis.
>
> ---
>
> **Question 1:** Learning from the fact that some architectures of ViT transfers better to some CNN architectures? e.g., Swin-B to VGG?
>
> Thank you for this insightful question, which prompts us to delve deeper into the transferability patterns between ViT and CNN architectures.
>
> As highlighted in prior work [1], ViTs and CNNs learn distinct feature representations, with notable differences particularly in their early-layer computations. Specifically, [1] demonstrated that aligning ResNet (a CNN) layers to match the hidden representations of a ViT requires modifying more layers than the reverse process (aligning ViT layers to match ResNet). This observation reveals an inherent asymmetry in cross-architecture alignment: ViTs are more adaptable to aligning with CNNs than CNNs are to aligning with ViTs.
>
> This asymmetry sheds light on why certain ViT architectures (e.g., Swin-B) transfer better to specific CNNs (e.g., VGG). ViTs, by nature, exhibit a stronger capacity to capture unique structural features—features that are generalizable enough to align with the representational spaces of CNNs. In contrast, CNNs, with their inductive biases toward local spatial hierarchies, are less flexible in adapting to the global, token-based features learned by ViTs. Consequently, when generating adversarial examples, ViTs can encode features that resonate more effectively with the internal representations of CNNs, leading to stronger transferability from ViT to CNN architectures.
>
> We appreciate this opportunity to clarify this mechanism, and we will emphasize this insight in the revised manuscript. Thank you again for your valuable feedback.
>
> [1] Do vision transformers see like convolutional neural networks? NeurIPS 2021

---

> > ### Comment · Reviewer_Qa5A · 2025-08-08
> > **Rebuttal Response**
> >
> > I thank the authors for their thorough rebuttal and thus, I maintain my positive rating.

---

> > > ### Author Response · Authors · 2025-08-08
> > >
> > > Thanks for your positive feedback on our work. We appreciate your valuable and constructive suggestions.

---

### Official Review · Reviewer_ZtAM · 2025-07-03

**Clarity:** 2
**Significance:** 3
**Originality:** 3
**Rating:** 4
**Confidence:** 4

**Summary:**

This paper proposes Spatial Adversarial Alignment (SAA), a method that improves adversarial transferability by improving upon the model alignment method. Model alignment increases the transferability of adversarial examples by fine-tuning a source model to align its predictions with those of a fixed witness model, using an alignment loss that captures prediction divergence. SAA introduces two improvement: a spatial-aware alignment that encourages alignment across both global and local regions, and an adversarial-aware alignment that focuses the alignment on adversarially perturbed inputs. Experiments on ImageNet show that SAA achieves improved transferability than the original model alignment approach.

**Questions:**

See weakness.

**Ethical Concerns:**

["NO or VERY MINOR ethics concerns only"]

**Final Justification:**

My main concern has been addressed by the authors during rebuttal, thus the score increase. However, I remain concerned about the quality of the revised paper given the numerous changes, clarifications, and additional details the authors have committed to include.

**Limitations:**

Although the paper briefly mentions computational complexity, it represents a major trade-off compared to the original MA method and deserves a more detailed discussion.

**Quality:**

2

**Strengths And Weaknesses:**

**Strengths**

1. Improving the transferability of adversarial perturbations is a practically relevant goal, and this paper builds on the recent model alignment framework. The proposed SAA method introduces two enhancements—spatial-aware and adversarial-aware alignment—that further improve adversarial transferability.

2. The main experiment in this paper is extensive, including comparison with model alignment on the Imagenet dataset. The ablation study demonstrate the compatibility with various existing transfer-improving methods. Table 7 also helps understand how each component of the SAA method contribute to the overall improvement.

**Weaknesses**

1. Spatial-alignment component of SAA requires clearer motivation and explanation of its formulation.
    1. Several claims in lines 48–61 require either citations or supporting evidence to justify the motivation. For instance, the paper mentions that cross-architecture transferability is limited because "features are not aligned in space" and that "the spatial features of ViTs and CNNs are different." Without references or empirical evidence, these statements feel speculative and make it difficult to understand the spatial alignment component.
    2. The claim that “MA only uses the final prediction of the model” is misleading. The alignment loss in Eq. (1) of Ma et al. can be applied to intermediate representations, and Table 4 in their paper explicitly explores alignment in the embedding space.
    3. Section 2.2, particularly the spatial-aware alignment portion, lacks clarity. Several parts (e.g., lines 104–105, 108–112, and 115–122) should include citations.
    4.  The use of the term “global features” to refer to logits is unconventional and potentially confusing. It is unclear why logits are considered “global” and how that definition helps in the context of alignment.
    5.  The explanation in lines 136–146 is difficult to follow due to unspecified dimensions. For example, what is the dimensionality of $z$? Assuming $H$ and $W$ are image dimensions, does having $H\times W$ local regions imply that each pixel is treated as a separate region? Clarifying these points would significantly improve the readability of the spatial alignment method.
    6. It remains unclear how the proposed local/global feature perspective addresses the cross-architecture misalignment problem raised earlier. A more explicit connection between motivation and method would strengthen the argument.

2. Section 2.3: It seems that “feature similarity” refers to logit similarity, but this should be clarified. Moreover, wouldn’t an increase in cosine similarity naturally be expected from any source model that has improved transferability? (i.e., unaligned model A vs unaligned model B, where perturbations from B has better transferablity.) Also, a comparison with MA could potentially better understand the improvement from SAA.

3. Computational overhead: The paper does not discuss the computational cost introduced by the two new alignment components. A quantitative analysis (e.g., number of additional forward/backward passes or runtime comparison) would be helpful for understanding the trade-offs compared to the original MA method.

---

> ### Author Rebuttal · Authors · 2025-07-31
>
> Thanks for your thoughtful suggestions. During the rebuttal period, **we have made careful efforts to address your concerns**. Additionally, we have provided detailed responses to your other questions. **We earnestly hope that you will reconsider our paper.**
>
> ---
>
> **Weakness 1.1:** Supports on SA.
>
> Thanks for your valuable feedback on the motivation in lines 48–61. The analysis in lines 48–61 focuses on potential reasons for the limited cross-architecture transferability of MA, and this is directly validated by the qualitative and quantitative experiments in Section 2.3—our core effort to verify whether SAA improves spatial and adversarial feature alignment.
>
> - For **quantitative evidence**: As detailed in Section 2.3, we use cosine similarity to measure global feature similarity (since spatial features here refer to entire image representations). Table 1 clearly shows two key results:
>   1. there is a significant inherent feature gap between diverse architectures (e.g., Res50 and Swin-B), confirming that spatial features of CNNs and ViTs are distinct;
>   2. SAA consistently reduces this gap (e.g., a 49.3% improvement for Res50-Swin-B on adversarial examples), demonstrating that unaligned spatial features hinder transferability—directly supporting the claim that "features are not aligned in space." These results empirically validate the existence of architectural feature differences and misalignment.
>
> - For **qualitative evidence**: Grad-CAM visualizations in Figure 2 illustrate that unaligned surrogates exhibit feature distributions distinct from witness models, while aligned surrogates (via SAA) learn more consistent patterns—confirming improved spatial alignment. Additionally, Appendix E provides further visualizations: adversarial examples from unaligned surrogates retain heatmap distributions similar to clean inputs (indicating poor transfer due to misaligned features), whereas aligned surrogates produce adversarial examples that disrupt witness model attention, validating the impact of spatial feature alignment on transferability.
>
>
> To address the need for citations, we will explicitly reference relevant literature in the revised version: "features are not aligned in space" (supported by PNA [51]), and "the spatial features of ViTs and CNNs are different" (supported by [1], [2]).
>
>
>
> ---
>
> **Weakness 1.2:** Final prediction.
>
> Thank you for pointing out this concern—we appreciate the opportunity to clarify our statement about Model Alignment (MA, ECCV 2024).
>
> Our claim that "MA only uses the final prediction of the model" is grounded in both MA’s official paper and its implementation, and we aim to clarify this alignment:
>
> 1. First, while MA’s Section 3.1 presents formalizations for alignment across different layers, their Section 3.2 explicitly states: "Specifically, when $q = l$, the alignment loss measures the divergence between the probability distributions generated by the two models. In this scenario, the Kullback-Leibler (KL) divergence is particularly suitable, due to its effectiveness in measuring distribution differences and its relative ease of implementation in practice", which aligns with our description of "using the final prediction-related representations" of the model.
>
> 2. Second, MA’s publicly available code (model-alignment/src/align.py, lines 35–37) explicitly uses the model’s final output as the basis for computing the alignment loss. This implementation directly confirms that MA focuses on the final prediction-related representations rather than intermediate layers in practice.
>
> Thus, our statement is consistent with MA’s own paper and official code. We welcome reviewers to verify this via their publicly accessible repository.
>
> ---
>
> **Weakness 1.3:** References on SA.
>
> We will add citations to MA [32] in lines 104–105 and 108–112, and to [2] in lines 115–122. In addition, we will further check the rigor of the description, add the corresponding citations, and update them in the final version.
>
> ---
>
> **Weakness 1.4:** Global features.
>
> Thank you for pointing this out. We would like to clarify that the term "global feature" is primarily used to distinguish it from local features in spatial-aware alignment. This term is used because, in the vision backbone, the final output features reflect global image information.
>
> We will further refine the description in the main body. First, we will highlight the definition and description of global and local features in lines 130-144. Then, we will further deepen the intuitive understanding of global and local features by combining them with Figure 1. We have provided code in the supplementary materials, which is consistent with the description in the paper and hopes to further bridge any gaps in understanding.
>
> ---
>
> **Weakness 1.5:** Dimension explanation.
>
> Thank you for pointing out this issue. The $H$ and $W$ here refer to the dimensions of the local features, not the image dimensions. In image classification models, the downsampling factor is typically 32. For a $224 \times 224$ image, downsampling results in local features of $(224/32)\times(224/32)=7\times 7$. In this case, $H=W=7$ so that the dimension of $z$ is $7 \times 7=49$. We will update this description in the final version.
>
> ---
>
> **Weakness 1.6:** Feature misalignment.
>
> Thank you for this insightful question, which prompts us to clarify the explicit connection between our motivation (addressing cross-architecture misalignment) and the proposed local/global feature perspective.
>
> As summarized in lines 48–61, the limited cross-architecture transferability of MA stems from its exclusive reliance on aligning surrogate and witness models via their final global outputs. While MA assumes feature similarity correlates strongly with transferability, its poor performance across architectures reveals a critical gap: relying solely on final global features fails to capture meaningful similarity in cross-architecture settings. This gap arises because CNNs and ViTs inherently differ in structural designs—including receptive fields (local vs. global), layer stacking mechanisms, and normalization techniques—leading to misaligned spatial and structural features that hinder transferability.
>
> Our proposed spatial-aware alignment directly addresses this by explicitly bridging local and global features, creating a clear link between motivation and method:
> - Beyond aligning final global features (as MA does), we additionally focus on learning local regional features. This dual focus targets the structural differences between CNNs and ViTs: local features capture fine-grained spatial patterns critical for CNNs, while global features retain the holistic token-based representations central to ViTs.
> - By integrating both perspectives, surrogate models learn not just high-level global similarity but also low-level structural nuances unique to each architecture. This enables them to acquire "unique structural features" that transcend architectural differences—features that are aligned both spatially (locally) and semantically (globally), thereby mitigating cross-architecture misalignment.
>
> In essence, the local/global feature perspective directly responds to the root cause of misalignment (structural differences between CNNs and ViTs) by ensuring alignment at both granularities, strengthening transferability where MA falls short.
>
>
> ---
>
> **Weakness 2:** Feature similarity.
>
> Yes, we have explained in line 188 that feature similarity is calculated using cosine similarity of global features. In Section 2.3, we focus on **verifying whether SAA improves the spatial and adversarial features of surrogate models after alignment**. Therefore, we need to verify the feature similarity before and after alignment and the corresponding visualization. Table 2 of the main body already fully demonstrates that the adversarial transferability brought by SAA significantly exceeds that of MA. Therefore, we focus more on the nature of SAA itself in this section. For more qualitative and quantitative analysis, please refer to **Weakness 1**.
>
> While it might be obvious that a model with better transferability can improve a surrogate model with poor transferability when used as a witness model, in SAA, a model with average transferability can further improve a surrogate model with better transferability. For example, when Res50 is aligned with ViT-B, Res50 learns unique structural features, allowing the generated adversarial examples to still improve transferability. SAA reveals this phenomenon and demonstrates that there are still many areas of urgent research in feature similarity and adversarial transferability.
>
> ---
>
> **Weakness 3 & Question 1:** Computational overhead.
>
> In "GPU Memory and Computing Cost" of Appendix C, we report the training time (GPU hours). For computing cost, training SAA for one epoch on ImageNet (with a batch size of 64) requires about 10 hours using an Nvidia RTX 3090.
>
> Here, we analyze two alignments: spatial-aware alignment supports parallelization, so only one forward and backward propagation is required for loss calculation; for adversarial-aware alignment, since adversarial examples need to be generated, $t$ forward and backward propagations are required. Therefore, the time complexity of training one epoch is $O(1+t)$. In the above practice, spatial-aware alignment and adversarial-aware alignment take about 2 and 8 hours respectively.
>
> Note that the main computational bottleneck is actually obtaining adversarial examples, not model training. Some newer adversarial example techniques [3] may be able to accelerate this process, which is a very promising research direction.
>
> [1] Do vision transformers see like convolutional neural networks?, NeurIPS 2022
>
> [2] Understanding the robustness in vision transformers, ICML 2022
>
> [3] Revisiting Adversarial Training at Scale, CVPR 2024

---

> > ### Comment · Reviewer_ZtAM · 2025-08-04
> >
> > Thank you very much for your detailed response. I have several remaining concerns:
> >
> > 1.1. A more thorough analysis is needed to support the claim that "the spatial features of CNNs and ViTs are distinct." For example, in Table 1, ViT-B shows higher similarity to DN121 (a CNN model) than to SWIN-B (a ViT model), which appears to directly contradict the claim.
> >
> > 1.2 "Specifically, ... in practice" reads as an example illustrating the interpretation of the alignment loss under this particular choice of parameters. As I stated in the original review, the results in Table 4 of their paper also contradicts the claim that "MA only uses the final prediction of the model".
> >
> > 1.4 "This term is used because, in the vision backbone, the final output features reflect global image information." I am a bit confused here. Are you referring to the logits or the output of the penultimate layer? Also, while I appreciate the effort to offer an intuitive distinction between global and local features, it would be helpful to include concrete empirical evidence demonstrating that intermediate and final representations capture localized and global semantics, respectively (or providing appropriate reference to it).

---

> > > ### Author Response · Authors · 2025-08-06
> > >
> > > **Question 1.1:** Spatial features of CNNs and ViTs are distinct.
> > >
> > > First, Figure 4 in Appendix E directly shows that the Grad-CAM heatmaps for the same image obtained by ResNet-101 and ViT-S are different, qualitatively demonstrating the difference in spatial features between CNN and ViT. Due to rebuttal limitations, we cannot provide further figures here. However, we promise to provide more visualizations in the final version.
> > >
> > > Then, we enhance the experiment in Table 1 (randomly sample 100 images from ImageNet val and then compute the cosine similarity between global features), as shown in the following:
> > >
> > > |Cos. Sim.|Res18|Res50|Res101|VGG19|IncV3|DN121|ViT-B|SWIN|PVT_v2|
> > > |:---------:|:----------:|:----------:|:----------:|:----------:|:----------:|:----------:|:----------:|:----------:|:----------:|
> > > |Res18| **1.0000** |0.9897|0.9876|0.1224|0.1040|0.1053|0.0687|0.0555|0.0583|
> > > |Res50|0.9897| **1.0000** |0.9921|0.0883|0.0910|0.0882|0.0566|0.0465|0.0514|
> > > |Res101|0.9876|0.9921| **1.0000** |0.0379|0.0674|0.0473|0.0475|0.0521|0.0311|
> > > |VGG19|0.1224|0.0883|0.0379| **1.0000** |0.4360|0.5606|0.2653|0.3753|0.3278|
> > > |IncV3|0.1040|0.0910|0.0674|0.4360| **1.0000** |0.4824|0.3323|0.3234|0.3191|
> > > |DN121|0.1053|0.0882|0.0473|0.5606|0.4824| **1.0000** |0.4016|0.3942|0.3651|
> > > |ViT-B|0.0687|0.0566|0.0475|0.2653|0.3323|0.4016| **1.0000** |0.3058|0.3980|
> > > |SWIN|0.0555|0.0465|0.0521|0.3753|0.3234|0.3942|0.3058| **1.0000** |0.6287|
> > > |PVT_v2|0.0583|0.0514|0.0311|0.3278|0.3191|0.3651|0.3980|0.6287| **1.0000** |
> > >
> > > This table provides clear insights into the feature relationships between CNNs and ViTs, revealing distinct features within and across architectural families:
> > >
> > >   **1. High Feature Similarity Within the Same Architectural Family**
> > >   - **ResNets (Res18, Res50, Res101)**: These CNNs exhibit extremely high cosine similarity (0.9876–0.9921), approaching 1.0. This indicates that when the spatial features of the model is highly consistent, the cosine similarity will be close to 1. While not as homogeneous as ResNets, other CNNs (VGG19, IncV3, DN121) show moderate intra-family similarity. For example, VGG19 and DN121 have a similarity of 0.5606, and IncV3 and DN121 share 0.4824, reflecting commonalities in local receptive fields and hierarchical pooling.
> > >
> > >   - **ViTs Variants (ViT-B, SWIN, PVT_v2)**: ViTs also display meaningful intra-family similarity, though lower than ResNets. SWIN and PVT_v2 show a high similarity of 0.6287, likely due to their shared patch-merging mechanisms and spatial-aware attention. ViT-B and PVT_v2 share 0.3980 similarity, indicating partial overlap in their global token-based feature aggregation.
> > >
> > >
> > >   **2. Low Cross-Architecture Similarity Between CNNs and ViTs**:
> > >   A striking pattern is the **weak feature alignment between CNNs and ViTs**, with most cross-type similarities ranging from 0.03 to 0.4, lower than average intra-family values.
> > >   - **ResNets vs. ViTs**: Res18/50/101 show extremely low similarity with all ViTs (0.0311–0.0687). This suggests that ResNets’ local-feature-driven, hierarchical representations differ fundamentally from ViTs’ global token-based features.
> > >   - **Other CNNs vs. ViTs**: Some CNNs (VGG19, DN121) exhibit slightly higher similarity with ViTs (e.g., VGG19-SWIN: 0.3753; DN121-ViT-B: 0.4016) compared to ResNets. This may stem from architectural differences: VGG19 uses simpler stacking of convolutions, and DN121 (DenseNet) emphasizes feature reuse, which might align more with ViTs’ global information flow.
> > >
> > >   **3. Key Implications**
> > >   - The large gap between CNNs and ViTs confirms that their core mechanisms (convolution vs. self-attention) lead to distinct feature spaces—CNNs prioritize hierarchical local patterns, while ViTs focus on global token relationships.
> > >   - High similarity within ResNets and moderate similarity within ViTs suggest that architectural homology (e.g., residual blocks for CNNs, attention mechanisms for transformers) fosters shared feature representations.
> > >   - The relatively higher similarity between VGG19/DN121 and ViTs hints that CNNs reduced inductive bias toward local feature hierarchies may have feature distributions closer to transformers.
> > >
> > >   In summary, the table shows that CNNs and ViTs form distinct spatial features. Table 7 also shows that the transferability is improved after aligning local features, which also verifies that the spatial features are not completely consistent before alignment.
> > >
> > > Finally, a lot of work has supported that the spatial features of CNNs and ViT are different: [A1] shows ViTs have more uniform representations across all layers than CNNs; [A2] examines the role of self-attention in learning robust representations and shows the ViT's features is different from CNNs.
> > >
> > > We will integrate the above into the final version. Thank you for your feedback.
> > >
> > > [A1] Do Vision Transformers See Like Convolutional Neural Networks? NeurIPS 2021
> > >
> > > [A2] Understanding The Robustness in Vision Transformers, ICML 2022

---

> > > ### Author Response · Authors · 2025-08-06
> > >
> > > **Question 1.2:** Final prediction.
> > >
> > > Thanks for your further clarification, which helps us address this point more precisely. To resolve this ambiguity, we will correct the corresponding description in the final version to "MA only uses the final prediction of the model in the official implementation," annotate the corresponding GitHub code snippet in the footnote, and explain the special case of Table 4.
> > >
> > > Except for Table 4, MA's experiments are based on the officially released code. Table 4 of MA is an ablation experiment, and its implementation differs from that of MA in the main body. Appendix B of MA describes how the implementation of Table 4 requires additional training of a feature alignment neural network.
> > >
> > > This revision will ensure our statement accurately reflects MA’s practical implementation while acknowledging the special case of their ablation study. Thank you again for helping us refine this clarification.
> > >
> > > ---
> > >
> > > **Question 1.4:** Global features and local features.
> > >
> > > Thanks for your further questions. We will respond to this issue from three aspects: **the definition of features**, **why we choose logits**, and **the improtance of spatial features**.
> > >
> > > 1. **We actually give the definition of "global features" and "local features" in the main body:**
> > >
> > >    - **Global features** are defined in **lines 130-132**: For ease of understanding, we define the global features $f_\theta(x)$ as the logits of the model corresponding to the input $x$.
> > >      - For CNNs, it is the output of features by the last layer.
> > >      - For ViTs, it refers to the final embedding of the $\texttt{[CLS]}$ token after the MLP block.
> > >
> > >    - **Local features** are defined in **lines 136-143**: Let $z_{\theta_s}^{[q]}(x)$ and $z_{\theta_w}^{[q]}(x)$ denote the local features associated with each local region $q$ for the surrogate model and witness model, respectively.
> > >      - For CNNs, $z_\theta(x)^{B\times C\times H \times W }$ represents the logits are generated by the final convolutional layer and then passed through the final MLP.
> > >      - For ViTs, $z_\theta(x)^{B\times C'\times H \times W }$ is the embeddings of patch tokens after passing through the last MLP except for the $\texttt{[CLS]}$ token, where each patch token corresponds to a specific spatial region in the input.
> > >
> > >     Thus, in this paper, **the global features are the model's logits for image $x$, and the local features are the model's logits for the local $z$ of the image.** The final features, after passing through the MLP, still retain global information, allowing us to perform image classification.
> > >
> > > 2. **The main reason we choose logits over features before passing through the MLP is to align the feature dimensions.** For example, ViT's patch embeddings $z(x)$ are $(B, N, C')$ by default. We first transform it to $(B, C', H', W')$, where $N=H'\times W'$. Then, we perform an adaptive pooling operation to transform it to $(B, C', H, W)$. The model's MLP ensures that the channels $C'$ are consistent (on ImageNet is 1000), which inherently guarantees minimal feature modification. For example, in MA embedding alignment (Table 4 of MA), since features cannot be guaranteed to have consistent channel dimensions, an additional feature alignment network is introduced, which requires additional training and cannot guarantee that the features passed through the network are aligned.

---

> > > ### Author Response · Authors · 2025-08-06
> > >
> > > **Question 1.4:** Global features and local features. (following the above comments)
> > >
> > >
> > > 3. **Many representative works have shown that intermediate and final representations capture localized and global semantics:**
> > >
> > >    - In the work written by Turing Award winners [B1], it is summarized in detail that CNN exploits the property that many natural signals are compositional hierarchies, in which higher-level features are obtained by composing lower-level ones. In images, local combinations of edges form motifs, motifs assemble into parts, and parts form objects. **This work shows that intermediate features in CNN can capture local semantics such as shape and texture, while the final representations represent the meaning of the object.**
> > >
> > >    - In ViT [B2], the intermediate representations themselves refer to patch tokens, and the final representation is the $\texttt{[CLS]}$ token, which is a relationship defined by the structure of ViT itself. The **[CLS] token** is initially a learnable parameter. After passing through the self-attention mechanism of the multi-layer Transformer encoder, it gradually aggregates the key information from all patch tokens, ultimately becoming a "compressed representation" of the entire image—encapsulating the overall semantics of the image (such as object category and scene attributes). Each **patch token** represents an abstract feature of the corresponding image region, representing local information about the image.
> > >
> > >    The main reason why introducing spatial-aware alignment further aligns features and improves adversarial transferability is that, **when using only global features for supervision, gradients from the final loss during backpropagation may significantly weaken as they propagate to shallow layers, leading to poor training of early layers, resulting in incomplete feature alignment.** After introducing local feature supervision, the intermediate layer’s outputs are explicitly aligned. During training, the total loss is a weighted sum of the local alignment loss and global alignment loss. This ensures that:
> > >    - Shallow layers receive direct supervision signals, avoiding gradient dilution.
> > >    - Intermediate layers are forced to learn discriminative features relevant to the target task, rather than becoming "redundant" in deep stacks.
> > >
> > >     **Grad-CAM visualizations in Figure 2** empirically illustrate that unaligned surrogates exhibit feature distributions distinct from witness models, while aligned surrogates (via SAA) learn more consistent patterns—confirming improved **spatial alignment**. The ablation study of Table 7 also verifies the improvement of local alignment in adversarial transferability.
> > >
> > > Thank you for taking the time to participate in the discussion. We will update the main body and appendix of the final version with the above content. Thank you again for your help in our work.
> > >
> > > [B1] Deep learning, Nature 2015
> > >
> > > [B2] An Image is Worth 16x16 Words: Transformers for Image Recognition at Scale, ICLR 2021

---

> > > ### Author Response · Authors · 2025-08-08
> > >
> > > Dear Reviewer ZtAM,
> > >
> > > Thank you again for your diligent efforts and valuable feedback. We have carefully addressed your main concerns in detail, and we hope you will find our response satisfactory, as with the feedback provided to the other reviewers. With the discussion period drawing to a close, we eagerly look forward to your further comments, if any. Please know that we would be happy to clarify any additional concerns you may have.
> > >
> > > Best regards,
> > > The Authors

---

> > > > ### Comment · Reviewer_ZtAM · 2025-08-09
> > > >
> > > > Thank you for the thorough response. My main concerns have been addressed, and I will adjust the score accordingly.

---

> > > > > ### Author Response · Authors · 2025-08-09
> > > > >
> > > > > We appreciate your positive feedback and the update on the score. We are also thankful for the multiple communication rounds, as your involvement has significantly improved our work quality.

---

### Official Review · Reviewer_uYz7 · 2025-07-03

**Clarity:** 3
**Significance:** 3
**Originality:** 2
**Rating:** 4
**Confidence:** 3

**Summary:**

This paper proposes Spatial Adversarial Alignment (SAA), a method for enhancing the transferability of adversarial examples across architectures, especially from CNNs to ViTs. The method leverages a witness model to fine-tune a surrogate model by aligning both global and local features (spatial-aware alignment) and adversarial representations (adversarial-aware alignment). Experiments show that SAA improves adversarial transferability significantly over prior model alignment (MA) methods and can enhance the performance of multiple transfer attacks under black-box and ensemble settings.

**Questions:**

How robust is SAA to witness model choice? If the witness model is very weak (e.g., shallow CNN), does alignment still help?

What is the compute cost of SAA training? How many GPU hours are needed per model alignment? This is especially relevant for large ViTs.

What happens if the surrogate and witness have no architectural overlap (e.g., MobileNet and Swin-B)? Does SAA still improve transferability?

How often do pseudo-labels differ from the true labels? Have you analyzed the reliability of local feature alignment via argmax over logits?

Why use KL divergence for local logits? Did the authors try feature-matching (e.g., MSE between embeddings), or attention-head alignment as in recent interpretability work?

In transfer settings where the target model is known (but frozen), wouldn’t it make more sense to align to it directly rather than via a witness?

**Ethical Concerns:**

["NO or VERY MINOR ethics concerns only"]

**Final Justification:**

In my opinion, the paper does contribute to the idea of Spatial Adversarial Alignment. The key work that needs to be address is the foundational differences between architectures like CNNs and ViTs and the issue around SAA. This would allow the paper to be a notch up.

**Limitations:**

Yes

**Quality:**

3

**Strengths And Weaknesses:**

Strengths

The paper presents an array of novel contributions and new ideas. SAA integrates spatial-level and adversarial-level feature alignment, going beyond prior work that focused only on logits or global outputs.

There are strong empirical results supported by extensive experiments across 6 CNNs and 4 ViTs show substantial gains over existing methods, especially under cross-architecture black-box settings.

SAA supports multiple attack types. Unlike many prior works that improve a single attack (e.g., I-FGSM), SAA shows gains when applied to multiple families: MI, DI, TI, PatchOut, GI, etc.

SAA significantly boosts attack transferability between very different model families (e.g., CNN → ViT), which is a difficult and under-addressed problem.

The framework is designed as a training-time plug-in that does not change the attack itself — this modularity is appealing to practitioners.

Weakenesses

The compute budget is not reported. Surrogate model retraining with adversarial alignment is costly. The paper lacks reporting on total training time, epochs, GPU hours, or number of adversarial examples generated.

The security implications are unexplored. By boosting black-box adversarial transferability, SAA may exacerbate real-world model vulnerabilities, however the paper doesn’t reflect on the responsible disclosure or ethical implications of this.

There appears to be an overreliance on cosine similarity as a metric: Feature similarity is evaluated only via cosine similarity, which may not correlate strongly with transferability. A deeper metric (e.g., centered kernel alignment or probing transfer success vs. feature similarity) would add insight.

Label correctness of adversarial examples is not verified. The method assumes that adversarial examples retain their semantic label under perturbation and alignment, but this is not validated with human or automatic checks.

The "pseudo-labeling" step for local features may introduce noise. Aligning local logits using pseudo-labels from a potentially miscalibrated witness model could misguide learning, but no analysis or ablation is presented.

Training the surrogate to mimic the witness may lead to overfitting, reducing diversity in learned features. There’s no metric to evaluate how independent or useful the aligned features are post-attack.

---

> ### Author Rebuttal · Authors · 2025-07-31
>
> **Weakness 1 & Question 2:** The compute budget is not reported (total training time, epochs, GPU hours, or number of adversarial examples).
>
>  We would like to clarify that the relevant information is already included in the paper, and we will consolidate it into the "Implementation Details" section in the final version for better accessibility. Specifically:
>
> 1. In "Training Epochs" of Appendix C, we note that all experiments in this paper use 1 training epoch. As described in Algorithm 1, the number of adversarial examples generated by SAA is in a 1:1 ratio with the training dataset. Additionally, we explore performance differences across multiple training epochs and find that compared to MA, SAA achieves higher transferability even with fewer epochs, and its transferability reaches a higher upper bound after multiple training rounds.
>
> 2. In "GPU Memory and Computing Cost" of Appendix C, we report the training time (GPU hours). For computing cost, training SAA for one epoch on ImageNet (with a batch size of 64) requires approximately 10 hours using an Nvidia RTX 3090.
>
> ---
>
> **Weakness 2:** The security implications are unexplored.
>
> We would like to clarify that the relevant information is already included in Lines 333-335 of the main body: Adversarial examples by SAA exhibit enhanced adversarial transferability, especially in cross-architecture capabilities. This poses a huge threat to the deployment of real-world applications. Simultaneously, it is also conducive to better evaluating their adversarial robustness.
>
>
> ---
>
> **Weakness 3:** Overreliance on cosine similarity.
>
> In Section 2.3, our core focus is to verify whether SAA improves the **spatial and adversarial features** of surrogate models after alignment. Regarding the use of cosine similarity: since spatial features here refer to representations of entire images (not localized regions), cosine similarity serves as an intuitive and widely adopted metric to quantify global feature similarity between models. As shown in Table 1 of the main body, two key observations emerge:
> 1. there is a significant inherent feature gap between diverse architectures (e.g., Res50 and Swin-B);
> 2. SAA consistently reduces this gap, with meaningful improvements (e.g., the 49.3% similarity gain for Res50-Swin-B on adversarial examples). These results quantitatively confirm that SAA effectively identifies shared features between distinct models, which in turn enhances adversarial transferability.
>
> Cosine similarity quantitatively demonstrates that SAA can improve the feature similarity between surrogate and witness models, proving the validity of this paper's hypothesis. Regarding adversarial transferability, a more direct comparison can be found in Tables 2 and 3, where ASR (Adversarial Success Rate) better demonstrates attack performance. Given the diversity of CNN and ViT models, the relationship between feature similarity and attack performance has been underexplored. SAA opens this door, making it a promising future research topic.
>
>
> ---
>
> **Weakness 4:** Label correctness.
>
> By definition, adversarial examples are crafted with human-imperceptible perturbations (under $l_ₚ$ norm constraints) that mislead model predictions while preserving the original semantic content. In the field of adversarial machine learning, this property is foundational: the perturbations are strictly bounded to ensure they do not alter the image’s meaning as perceived by humans. This is a well-established convention, as $l_ₚ$-constrained perturbations (e.g., $ε = 16/255$ for $l_∞$ setting) are designed to be visually negligible and thus do not interfere with human recognition of the semantic label.
>
> Given this standard, verifying individual adversarial examples via manual or automatic checks for semantic label preservation is generally unnecessary in the literature, as the perturbation bounds inherently guarantee that the original label remains semantically valid. Our work adheres to this convention, with perturbations constrained to lₚ norms that ensure such preservation.
>
> ---
>
> **Weakness 5 & Question 4 & Question 5:** Pseudo-labels' noises.
>
> We should note that local pseudo-labels are not directly related to true labels because we learn local features of the surrogate and witness models, not global features.
>
> Furthermore, we add experiments using soft labels (logits) and the KL divergence loss function for local alignment, as shown below. The surrogate model is Res50 and the witness model is ViT-B. In this scenario, using hard labels (local pseudo-labels) achieves higher adversarial transferability (improving ASR by 6.1%).
>
>
> |          Attack         | Res18 | Res50 | Res101 | VGG19 | DN121 | IncV3 | ViT-B |  SWIN | PVT_v2 | MobViT | Avg. ASR (%) |
> |:-----------------------:|:-----:|:-----:|:------:|:-----:|:-----:|:-----:|:-----:|:-----:|:------:|:------:|:------------:|
> |      SAA-MI (ours)      | 84.1  | 99.6  |  74.7  | 80.3  | 81.8  | 65.7  | 24.3  | 48.7  |  52.9  |  62.5  |     63.9     |
> | SAA-MI (soft label, KL) | 81.9  | 96.4  |  65.9  | 74.5  | 77.7  | 60.8  | 18.8  | 44.4  |  43.1  |  53.0  |     57.8     |
>
> Finally, we include experiments with different distance metrics (which can be understood as embedding metrics) in the "Distance Metric" section of Appendix C. Here, we select Kullback–Leibler Divergence (KL), Total Variation (TV), Jensen-Shannon Divergence (JS), and NCE loss for evaluation. We select ResNet-50 as the surrogate model and ViT-B as the witness model for ablation studies. As shown in Table 8, KL exceeds the next best JS by 2.5%, so we choose KL as the metric.
>
> ---
>
> **Weakness6:** Overfitting the witness.
>
> The ultimate metric for evaluating the effectiveness of aligned features is the average attack success rate on the black-box model, not the accuracy of the surrogate model itself on ImageNet. In this paper, our alignment was trained for only one epoch, making it less susceptible to overfitting. Secondly, in "Training Epochs" of Appendix C, we further explore performance differences across multiple training epochs. While the transferability of MA converges rapidly with increasing epochs, SAA continues to improve until the ninth epoch. This indicates that MA easily overfits to the witness model, limiting its adversarial transferability. SAA, on the other hand, is less susceptible to overfitting, and its transferability reaches a higher upper bound after multiple training rounds, demonstrating the importance of using spatial and adversarial features for model alignment.
>
>
> ---
>
> **Question 1 & Question 3:** Witness model choice.
>
> In Section 3.2 of the main body, we have conducted a large number of experiments to demonstrate that SAA is robust to model selection. Although we conduct 16 combinations and evaluate on 6 CNNs and 4 ViTs, showing that SAA’s improvement is not limited to specific choices of witness models. Based on the above experiments, we further summarize the empirical guidance for the selection of surrogate and witness models:
> 1. Self-alignment (surrogate and witness models are consistent) usually only provides minor improvements (it is difficult to learn unique structural features), and alignment between different models improves more significantly.
> 2. When the surrogate model is a ViT-like model and the witness model is a CNN-like model, better cross-architecture transferability can be obtained. This is mainly because ViT has stronger performance, while CNN further provides unique structural features.
>
> If the shallow CNN is a well-trained model (for example, with normal performance on ImageNet), then there can be an improvement. Even when the surrogate model and witness model have no architectural overlap, SAA still improves adversarial transferability. As shown in Table 2 of the main body, when the surrogate model is DN121 and the witness model is Swin-B, SAA improves the average ASR by 10.9% and 12.6% compared to MI and MA, respectively.
>
>
>
> ---
>
> **Question 6:** Known target models.
>
> We should clarify that in a transfer-based black-box setting, no details of the target model are known. We chaim this in Lines 284-286: It should be noted that the target models are unknown in the transfer attack, so no information about the target models can be used for alignment. Therefore, your assumption is invalid. If the target model is known (but frozen), then the target model can be directly selected as the surrogate model, which is equivalent to a white-box attack.

---

### Decision · Program_Chairs · 2025-09-17

**Decision:**

Accept (poster)

**Comment:**

This work proposes Spatial Adversarial Alignment (SAA) as a means to boost transferability of adversarial examples across different DNN models and architectures. First, using a witness model and surrogate model, SAA promotes spatial alignment by minimizing divergence of features between the two models. Second, SAA uses a “self-adversarial” strategy that then realigns features to adversarial examples. Intuitively, this promotes attacks that focus on shared features of the models, that is, attack transfer.

Several weaknesses were highlighted, particularly: request for reporting of compute budget (authors clarified this was included), ethical impact of the work (while the attacks improve transferability, I don’t believe the work recognizes new attack surfaces or fundamentally new vulnerabilities; although the authors’ response doesn’t point to lines in the paper addressing this, and I’d urge the authors to include a clearer impact statement on the importance of highlighting security vulnerabilities in AI, and that while the paper improves existing approaches, the vulnerabilities highlighted are not fundamentally new to service providers), a range of clarifying remarks by ZtAM (I believe adequately responded to by the authors, in responses that will be quite feasible to roll in to the final version of the paper), a concern that the final paper will require a range of revisions (though the corresponding reviewer also indicated that the rebuttal and discussion adequately addressed concerns).

Further helpful notes were raised, that I didn’t consider to be significant weaknesses: reliance on cosine similarity (I didn’t view a strong dominating alternative), verification of labels (this isn’t common practice/a requirement for the work to have impact though nice to have), noise introduced by pseudo-labelling (the empirical evaluation in a sense circumvents this risk as the performance is viewed end-to-end), a desire for more qualitative evidence (while this was acknowledged as a limitation by the authors, I don’t think this is a key negative of the work nor community expectation in the area), and discussion of alignment vs direct ensemble model attacks (clarified by authors as different in terms of the approach taken though having similar objectives).

The reviewers observed the following main initial strengths: the integration of two approaches to feature alignment, comprehensive evaluation including 10x models of various kinds and different threat model settings including support for multiple attacks, ablations. While the paper was perhaps marginal in terms of ratings, I believe overall there are good merits for the paper supporting acceptance. There is conceptual and technical novelty (even if not technical depth which is not a requirement of impact) that is well aligned with the task of transferability. The results speak for themselves, it is apparent that the SAA approach has merit and is likely to have good impact to progress attack research where fundamentally new ideas are relatively rare now. It is particularly promising that this can support attacks generally, and overall the approach is quite practical/versatile.